

# A previously overlooked, highly diverse early Pleistocene elasmobranch assemblage from southern Taiwan

Chia-Yen Lin[1,*], Chien-Hsiang Lin[1,*] and Kenshu Shimada[2,3]

[1] Biodiversity Research Center, Academia Sinica, Taipei, Taiwan
[2] Department of Environmental Science and Studies and Department of Biological Sciences, DePaul University, Chicago, IL, United States of America
[3] Sternberg Museum of Natural History, Fort Hays State University, Hays, KS, United States of America
[*] These authors contributed equally to this work.

## ABSTRACT

The Niubu fossil locality in Chiayi County, southern Taiwan is best known for its rich early Pleistocene marine fossils that provide insights into the poorly understood past diversity in the area. The elasmobranch teeth at this locality have been collected for decades by the locals, but have not been formally described and have received little attention. Here, we describe three museum collections of elasmobranch teeth ($n = 697$) from the Liuchungchi Formation (1.90–1.35 Ma) sampled at the Niubu locality, with an aim of constructing a more comprehensive view of the past fish fauna in the subtropical West Pacific. The assemblage is composed of 20 taxa belonging to nine families and is dominated by *Carcharhinus* and *Carcharodon*. The occurrence of †*Hemipristis serra* is of particular importance because it is the first Pleistocene record in the area. We highlight high numbers of large *Carcharodon carcharias* teeth in our sample correlating to body lengths exceeding 4 m, along with the diverse fossil elasmobranchs, suggesting that a once rich and thriving marine ecosystem in an inshore to offshore shallow-water environment during the early Pleistocene in Taiwan.

## INTRODUCTION

The Indo-West Pacific is regarded as one of the crucial marine biodiversity hotspots in the world (*Myers et al., 2000*; *Bellwood & Meyer, 2009*). Most of the species are concentrated in the coral reef triangle area that has its northern limit extending to southern Taiwan. A remarkable 181 chondrichthyan species have been recorded in the modern fish fauna of Taiwan (*Ebert et al., 2013*), approximating 15% of the total number of global chondrichthyan species (*Weigmann, 2016*). Such species diversity is regarded as one of the highest biodiversity hotspots for elasmobranchs when considering the size of Taiwan (*Ebert et al., 2013*). However, how this remarkable chondrichthyan fauna was formed and evolved in the past are not well understood, primarily because relevant fossil records are traditionally overlooked or unstudied, despite being well-represented in the marine

Corresponding author
Chien-Hsiang Lin,
chlin.otolith@gmail.com

deposits of Taiwan. Thus, comparisons for associated fossil fauna and past biogeographic distributions are limited, particularly in the tropical-subtropical Pacific. *Lin et al. (2021)* highlighted the need for paleontological data for understanding the historical context of fish fauna and further recommended research potentials in the region.

In the Western Foothills of Taiwan, numerous Neogene to Quaternary strata are known to be rich in marine fossils (*e.g.*, *Ribas-Deulofeu, Wang & Lin, 2021*; *Lin & Chien, 2022*; *Lin & Chien, 2022*). For instance, the early Pleistocene Liuchungchi Formation in the Niubu area, Chiayi County, southwestern Taiwan is of particular research interest due to its abundance and diversity of marine fauna. This fauna includes mollusks (*Hu, 1989*; *Xue, 2004*), crabs (*Hu, 1989*; *Hu & Tao, 1996*; *Hu & Tao, 2004*; *Xue, 2004*), sea urchins (*Hu, 1989*; *Xue, 2004*), whale barnacles (*Buckeridge, Chan & Lee, 2018*), teleost bones (*Tao, 1993*) and otoliths (*Lin et al., 2018*), and elasmobranch teeth (*Xue, 2004*). Fossils from this region were collected by the late W.-J. Xue during the 1980s–2000s, and currently this large and diverse collection (over 3,000 specimens) is mainly deposited in the Chiayi Municipal Museum, Chiayi City, Taiwan (CMM). There is a considerable number of elasmobranch teeth from Xue's collection that were reported by *Xue (2004)* in the form of photographic atlas without descriptions, and another collection donated by Prof. Hsi-Jen Tao (National Taiwan University) to the Biodiversity Research Museum, Academia Sinica, Taipei, Taiwan (BRMAS) is available. An additional small collection is also deposited in the National Taiwan Museum (NTM). The purpose of this present study is to properly document the occurrences of these elasmobranch fossils from the Liuchungchi Formation at the Niubu locality based on these collections and a few newly collected specimens. The diverse association of teeth provides opportunities for obtaining a more complete view of the Pleistocene elasmobranch fauna in the rarely explored subtropical West Pacific.

## Geological setting

Since the late Miocene, the island of Taiwan was gradually uplifted by the Penglai orogeny—the collision between the Chinese continental margin and the Luzon Arc—and, subsequently, a series of subsiding foreland basins were formed in western Taiwan (*Ho, 1976*; *Suppe, 1984*; *Lundberg et al., 1997*; *Lin & Watts, 2002*; *Nagel et al., 2013*; *Chen, 2016*). These foreland basins gradually developed from north to south accumulating clastic sediments (*Ho, 1967*; *Covey, 1984*; *Teng, 1990*), and in the south the basins have high deposition rates (700–900 m/Ma) due to a deeper depositional environment (*Chen, Huang & Yang, 2011*). Thus, the depositional sequences reflect sea-level changes during the Quaternary that followed the 100 ky orbit eccentricity cycles (*Chen, Huang & Yang, 2011*; *Chen, 2016*). Meanwhile, thick pre-orogenic and synorogenic sediments infilling the foreland basin were squeezed and uplifted, which formed the 7–9 km Miocene to Pleistocene strata in the Western Foothills (*Yu & Chou, 2001*; *Nagel et al., 2013*).

The Liuchungchi Formation in the Niubu area, Chiayi County is exposed along the Bazhang River (Fig. 1B). Four successive formations are exposed from east to west: the Liuchungchi, Kanhsialiao, Erhchungchi, and Liushuang formations (*Stach, 1957*; *Chou, 1975*; *Chen, Huang & Yang, 2011*; *Chen, 2016*; Figs. 1B and 1C). The age of the Liuchungchi Formation is 1.90–1.35 Ma (*Chen, 2016*), with a deposition rate of about 700 m/Myr in

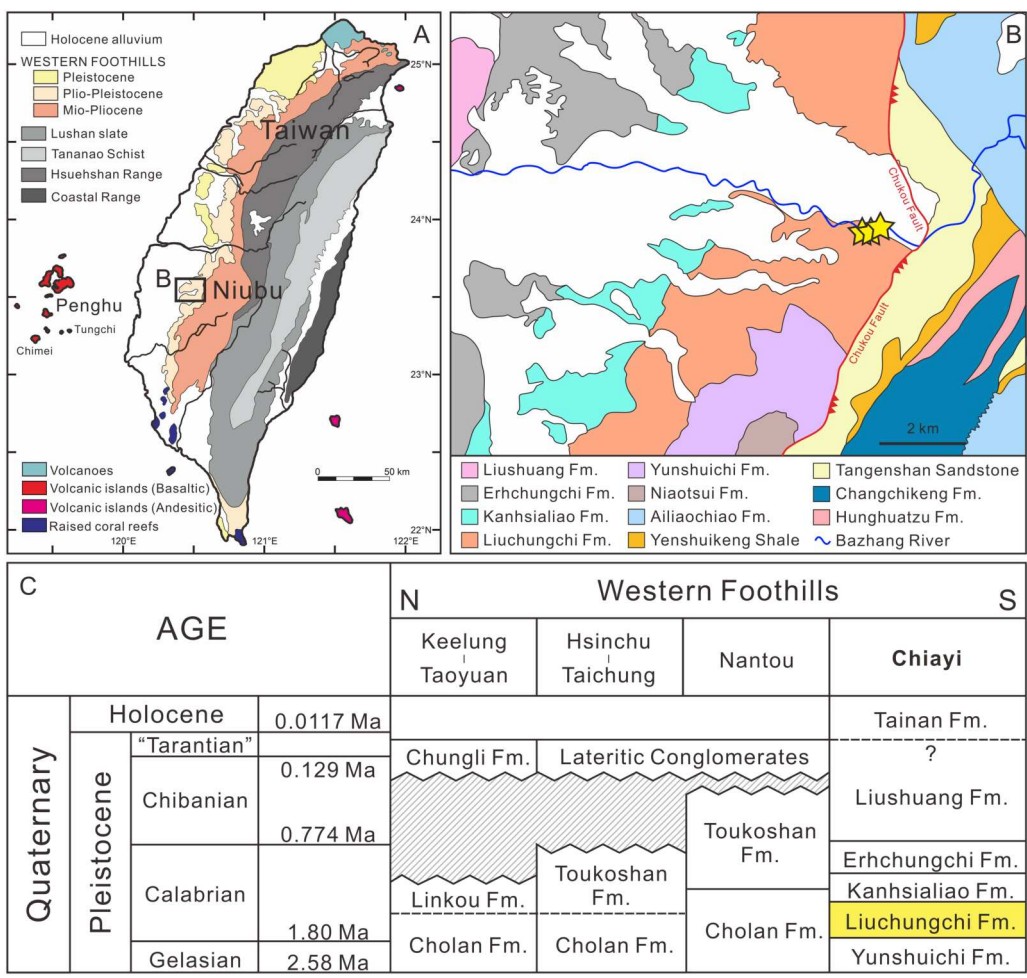

**Figure 1** **Summary of the sampling sites.** (A) Overview of geological map of Taiwan (modified after *Chen, 2016*). (B) Geological map of Nuibu area, Chiayi (map extracted from National Geological Data Warehouse, Central Geological Survey, MOEA). Yellow stars = sampling sites (see Fig. 2B for details). (C) stratigraphic correlation of the Western Foothills (modified after *Chen, 2016*). Liuchungchi Formation is indicated in yellow.

the lower section and 1,100 m/Myr upsection, the maximum thickness of the formation is 760 m (*Chen, Huang & Yang, 2011*). The Liuchungchi Formation is composed of dozens of depositional sequences, each representing a 41 ky climate cycle (*Chen, Huang & Yang, 2011*; *Chen, 2016*). The depositional environment can be divided into two distinct sections, with the lower sequence composed of thick sandstone with cross bedding, parallel bedding, and strong bioturbation reflecting shoreface to the offshore transition zone, and the upper sequence composed of interbeds of sandstone and shale and storm deposits in the form of sandstone, indicating the inner offshore (*Chen, Huang & Yang, 2011*; *Chen, 2016*).

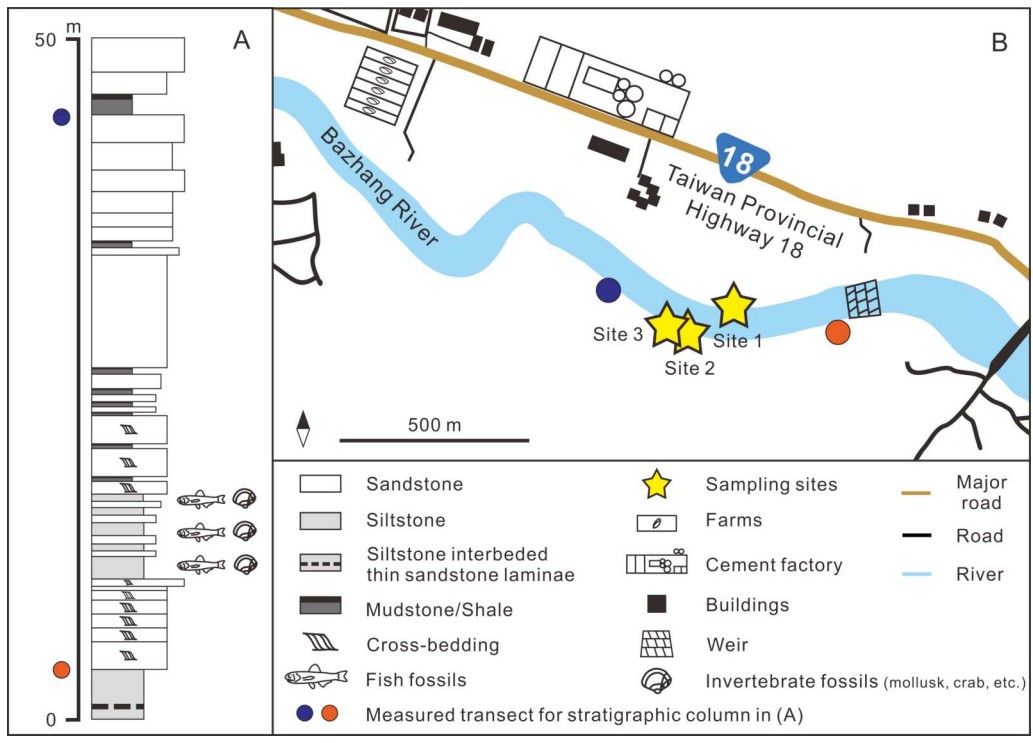

**Figure 2** (A) Stratigraphic column (modified after *Huang, 2010*); (B) details of the sampling sites. GPS coordinates: Site 1 = 23°26′23.4″N, 120°35′35.5″E; Site 2 = 23°26′22.6″N, 120°35′32.7″E; Site 3 = 23°26′23.5″N, 120°35′29.8″E.

## MATERIALS & METHODS

The fossil site is located in the Niubu area, Chiayi County, southwestern Taiwan, about 15 km east of Chiayi City (Fig. 1A). The layers containing fossils are exposed along the Bazhang River, just downstream of a dam near a high-voltage tower, where they are readily accessible during the dry seasons (winter) when the water level is low (Fig. S1). Fossil mollusks are very abundant in several of the condensed layers, as well as fragments of crabs, sea urchins, and teleost fish bones (Fig. 2A). Fossil shark teeth are rare based on both surface collecting and bulk sampling conducted during our several field trips between 2018–2022. Bulk sediment samples of over 830 kg (Sites 1–3 in Fig. 2B) were sieved (500-μm mesh) from the loosely cemented siltstone, yielding a large number of otoliths (Lin et al., unpublished data), but only one shark tooth and two ray teeth. We note the discrepancy in the numbers of elasmobranch specimens in museum collections and our field surveys, which can be explained by the fact that the larger sample sizes in museum collections primarily reflect collecting based on chance occurrences of shark teeth over the past 3–4 decades, compared to collecting based on our limited number of field surveys. Moreover, the initial purpose of the bulk sampling was for collecting teleost otoliths instead of elasmobranch teeth due to their abundance.

The upstream area of the Bazhang River contains strata older than the Liuchungchi Formation. They include the Neogene Tangenshan Sandstone and Yenshuikeng Shale, which are exposed approximately 1 km east of a weir (Figs. 1 and 2), with an elevation of more than 300 m above the level of our sampling sites. Both stratigraphic units are composed of consolidated sandstones with some marine fossils such as mollusks, which are different from the fine, unconsolidated siltstones of the Liuchungchi Formation. After storms and rainy seasons, numerous blocks of sandstone from these older strata can be found along the riverbed of the Bazhang River (Fig. S1). These blocks are lithologically distinct and confined to areas below our sampling sites. Although we consider our material as early Pleistocene and that the mixing of older Neogene fossils with our specimens is improbable, we cannot entirely rule out that certain worn material (*e.g.*, Myliobatiformes) is not reworked. Further geochemical analysis (*e.g.*, Sr-isotope, *Kocsis et al., 2021*; *Kocsis et al., 2022*) would help determine the extent of the reworking.

The BRMAS, CMM, and NTM collections analyzed here were collected from the surface exposures of the Niubu locality without bulk-sampling of sediments; however, the exact stratigraphic horizons and detailed lithology within the Liuchungchi Formation for each specimen are not known. Stacked images of teeth were taken and measurements of crown height (CH), mesial crown edge length (MCL), and basal crown width (BCW) were noted whenever possible. Specimens from the BRMAS are registered under ASIZF, CMM under CMM F, and those in the NTM are under NTM I. Because the Pleistocene is relatively close to modern times, the morphology of elasmobranch teeth has not changed much from that time to the present. Therefore, identifications of these fossil teeth were conducted by comparing them with teeth of extant taxa.

The diversity of our elasmobranch assemblage was compared with other Pleistocene assemblages to highlight its significance within the associated spatio-temporal context. Taxonomic composition and abundance data from the early Pleistocene temperate assemblages of Japan (*Karasawa, 1989*; *Kawase & Nishimatsu, 2016*; *Tanaka & Taru, 2022*) and tropical records from Java (*Koumans, 1949*; *Yudha et al., 2018*) and Sulawesi (*Hooijer, 1954*) were compared. We calculated diversity indices, including species richness (number of species recorded), Shannon's entropy (*Shannon, 1948*), Simpson's diversity index (*Simpson, 1949*), and Fisher's alpha (*Fisher, Corbet & Williams, 1943*) for a general comparison.

## Systematic paleontology

A summary of taxa and their numeric abundance are listed in Table 1. The elasmobranch assemblage contains 697 teeth, consisting of nine families and 20 taxa. The classification scheme follows that of *Nelson, Grande & Wilson (2016)*, except for the family Galeocerdonidae, which we follow *Fricke, Eschmeyer & Van der Laan (2022)*. General morphological terminology follows that of *Compagno (1984)*, *Compagno (2002)*, *Purdy et al. (2001)*, *Shimada (2002)*, *Purdy (2006)*, *Cappetta (2012)*, and *Ebert et al. (2013)*. The synonymy list is limited to relevant records from Taiwan (*Huang, 1965*; *Uyeno, 1978*; *Hu & Tao, 1993*; *Xue, 2004*; *Tao & Hu, 2008*).

**Table 1  Elasmobranchs from the early Pleistocene Liuchungchi Formation of Niubu, southern Taiwan.**

| Order | Family | Taxon | ASIZF | CMM | NTM | Total |
|---|---|---|---|---|---|---|
| Lamniformes | Carchariidae | *Carcharias taurus* | 1 | 1 | | 2 |
| Lamniformes | Lamnidae | *Carcharodon carcharias* | 28 | 25 | 2 | 55 |
| Lamniformes | Lamnidae | *Isurus oxyrinchus* | 4 | 1 | 1 | 6 |
| Carcharhiniformes | Hemigaleidae | †*Hemipristis serra* | 3 | 3 | 1 | 7 |
| Carcharhiniformes | Carcharhinidae | *Carcharhinus altimus* | 5 | 10 | 2 | 17 |
| Carcharhiniformes | Carcharhinidae | *Carcharhinus amboinensis* | 3 | 2 | | 5 |
| Carcharhiniformes | Carcharhinidae | *Carcharhinus leucas* | 17 | 53 | 1 | 71 |
| Carcharhiniformes | Carcharhinidae | *Carcharhinus limbatus* | 16 | 21 | 3 | 40 |
| Carcharhiniformes | Carcharhinidae | *Carcharhinus longimanus* | 18 | 16 | 2 | 36 |
| Carcharhiniformes | Carcharhinidae | *Carcharhinus obscurus* | 9 | 15 | 1 | 25 |
| Carcharhiniformes | Carcharhinidae | *Carcharhinus plumbeus* | 8 | 42 | 1 | 51 |
| Carcharhiniformes | Carcharhinidae | *Carcharhinus sorrah* | 1 | 10 | | 11 |
| Carcharhiniformes | Carcharhinidae | *Carcharhinus tjutjot* | 5 | 14 | | 19 |
| Carcharhiniformes | Carcharhinidae | *Carcharhinus* spp. | 88 | 110 | 10 | 208 |
| Carcharhiniformes | Carcharhinidae | *Rhizoprionodon acutus* | 2 | 6 | | 8 |
| Carcharhiniformes | Galeocerdonidae | *Galeocerdo cuvier* | 1 | 5 | 1 | 7 |
| Carcharhiniformes | Sphyrnidae | *Sphyrna lewini* | | 2 | | 2 |
| Myliobatiformes | Dasyatidae | Dasyatidae indet. | 2 | | | 2 |
| Myliobatiformes | Aetobatidae | *Aetobatus* sp. | 32 | 22 | 4 | 58 |
| Myliobatiformes | Myliobatidae | *Myliobatis* sp. | 9 | 20 | 1 | 30 |
| Indet. | Indet. | Indet. | 25 | 12 | | 37 |
| | Total | | 277 | 390 | 30 | 697 |

Class Chondrichthyes *Huxley, 1880*
Order Lamniformes *Berg, 1958*
Family Carchariidae *Müller & Henle, 1838*
Genus *Carcharias Blainville, 1816*
*Carcharias taurus Rafinesque, 1810*
(Fig. 3)
1978 *Odontaspis* sp.; Uyeno, pl. 1, Fig. 5.

**Referred specimens:** $n = 2$: ASIZF0100320, CMM F0204.

**Description:** CH = 12.92–16.83 mm; MCL = 12.18–15.54 mm; BCW = 7.07–7.84 mm. The teeth are characterized by a slender, dagger-like main cusp and a single pair of small lateral cusplets. The crown exhibits no serrations. The lingual protuberance of the root is prominent.

**Remarks:** The teeth of *Carcharias taurus* are similar to those of *Odontaspis noronhai* and *O. ferox* by having a slender main cusp and lateral cusplet. However, the lateral cusplets of *Odontaspis* are more pronounced than those of *C. taurus*, including the fact that teeth of *O. ferox* typically exhibit multiple pairs of lateral cusplets.

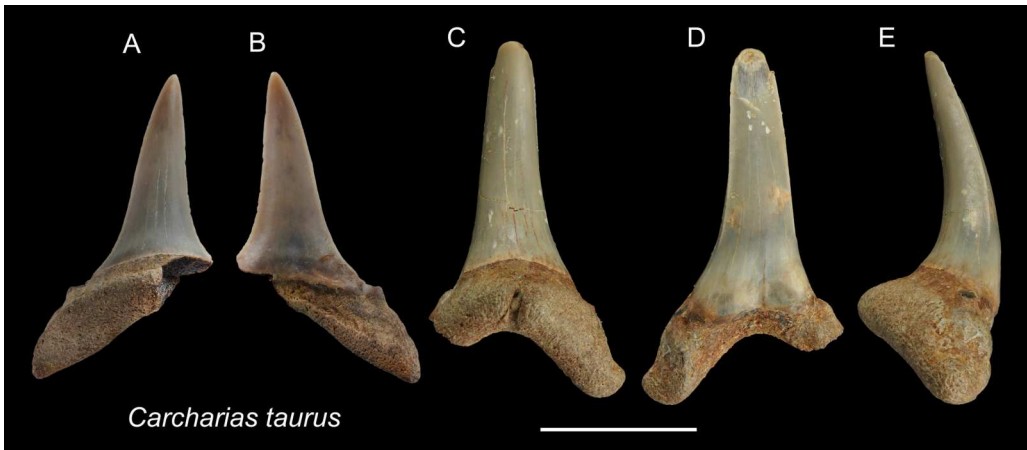

**Figure 3** **Teeth of *Carcharias taurus* from the early Pleistocene, Liuchungchi Formation of Niubu, southern Taiwan.** (A and B) ASIZF0100320; (C–E) CMM F0204. (A and C) lingual views; (B and D) = labial views; (E) lateral view. Scale bar = 1 cm.

Family Lamnidae *Bonaparte, 1835*
Genus *Carcharodon Smith, 1838*
*Carcharodon carcharias Linnaeus, 1758*
(Fig. 4)
1978 *Carcharhinus* sp.; Uyeno, pl. 1, Fig. 4, pl. 2, Fig. 7.
1978 *Carcharodon carcharias*; Uyeno, pl. 3, Figs. 12 and 13.
2004 Elasmobranchii indet.; Xue, pl. 1, Fig. 1–6, pl. 2, Fig. 1–7, pl. 3, Fig. 1, 2–7, pl. 7 Fig. 2.

**Referred specimens:** $n = 55$: ASIZF0100322–0100346, 0100435, 0100465, 0100530. CMM F0001–F0005, F0007–F0010, F0012–F0022, F0210, F0212, F2824, F2825, F2830, NTM I01122, I01123.

**Description:** CH = 6.76–41.03 mm; MCL = 9.61–45.68 mm; BCW = 8.74–37.09 mm. The upper teeth (Figs. 4A–4N) are broad and triangular. The cutting edge of both mesial and distal sides is almost straight with coarse serrations. The labial face of the crown is flat and the lingual face is convex, the crown is erect and symmetric to slightly distally inclined depending on tooth positions. The root is slightly arched, and the nutritive foramina and transverse groove are not prominent or absent. The lower teeth (Figs. 4O–4X) have a more robust but narrower serrated crown and bilobate roots with a rounded lingual face compared to the upper teeth.

**Remarks:** The genus *Carcharodon* is represented by three species: †*C. hastalis*, †*C. hubbelli*, and *C. carcharias*. †*Carcharodon hastalis*, which was traditionally placed in the genus *Isurus* or †*Cosmopolitodus*, lived through the Miocene and early Pliocene, †*C. hubbelli* in the late Miocene, and *C. carcharias* in the early Pliocene–Recent form a single lineage of chronospecies by developing serrations on their teeth (*Ehret et al., 2012*). The specimens described in this present paper exhibit well-developed serration consistent with teeth of †*C.*

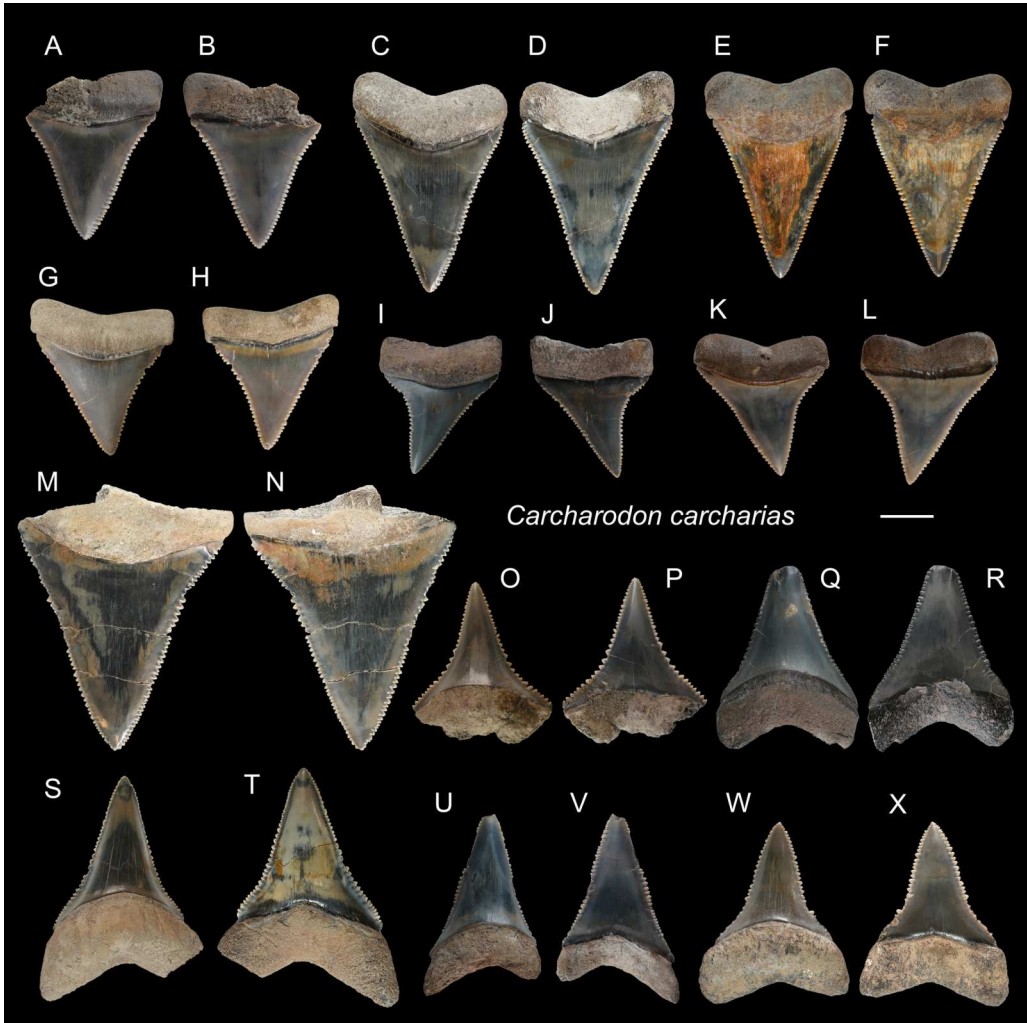

**Figure 4** **Teeth of *Carcharodon carcharias* from the early Pleistocene, Liuchungchi Formation of Ni-
ubu, southern Taiwan.** (A and B) ASIZF0100344; (C and D) ASIZF0100337; (E and F) ASIZF0100338; (G
and H) ASIZF0100336; (I and J) ASIZF0100335; (K and L) ASIZF0100339; (M and N) ASIZF0100340; (O
and P) ASIZF0100324; (Q and R ASIZF0100328; (S and T) ASIZF0100325; (U and V) ASIZF0100326; (W
and X) ASIZF0100323. (A, C, E, G, I, K, M, O, Q, S, U and W) lingual views; (B, D, F, H, J, L, N, P, R, T, V
and X) labial views. Scale bar = 1 cm.

*carcharias* (*e.g.*, *Hubbell, 1996*), and not like the teeth of †*C. hubbelli* with weak serrations
(*Ehret et al., 2012*). They include the largest dental remains among all the shark tooth
specimens described in this paper.

Genus *Isurus Rafinesque, 1810*
*Isurus oxyrinchus Rafinesque, 1810*
(Fig. 5)
?1965 *Isurus hastalis*; Huang, pl. 22, Figs. 12–14.
1993 *Isurus hastalis*; Hu & Tao, pl. 24, Figs. 6 and 8.
2004 Elasmobranchii indet.; Xue, pl. 5, Fig. 3, pl. 8, Fig. 4.
2008 *Isurus* sp.; Tao & Hu, pl. 2, Fig. 1–2.

**Referred specimens:** $n = 6$: ASIZF0100317–0100319, 0100321, CMM F0242, NTM I01131_1.

**Description:** CH = 9.86–27.81 mm; MCL = 12.21–26.89 mm; BCW = 9.31–9.96 mm. The anterior teeth have a slender, dagger-like, unserrated crown that is erect or lingually curved with an apical labial flexure (Figs. 5A–5H). The root, if preserved, has two rather narrow lobes with a moderately tight basal concavity. The lateral teeth have a flatter and broader, distally curved, unserrated crown with a short but mesiodistally wide root (Figs. 5I and 5J).

**Remarks:** Two extant species of *Isurus* are known: *I. oxyrinchus* and *I. paucus*. *Isurus oxyrinchus* has a more elongated and more labially curved crown than *I. paucus* (*Whitenack & Gottfried, 2010*). The teeth of *I. oxyrinchus* are also similar to those of *Carcharias taurus*, but the teeth of *C. taurus* have a pair of lateral cusplets that is absent in the teeth of *I. oxyrinchus* (*Wilmers, Waldron & Bargmann, 2021*). *Huang (1965)* reported a tooth of †*I. hastalis* (= *Carcharodon hastalis*; see above) from the Pleistocene Cholan Formation in Hsinchu, northern Taiwan; however, this species identification is questionable and the whereabouts of the specimen is unknown for verification.

Order Carcharhiniformes *Compagno, 1973*
Family Hemigaleidae *Hasse, 1878*
Genus *Hemipristis Agassiz, 1833–1843*
†*Hemipristis serra Agassiz, 1833–1843*
(Fig. 6)
1978 *Hemipristis serra*; Uyeno, pl. 1, Fig. 2.
2004 *Hemipristis* sp.; Xue, pl. 5, Figs. 1, 2, 5, 6 and 7.
2004 Elasmobranchii indet.; Xue, pl. 5, Fig. 5, pl. 7, Fig. 3, 5, pl. 9, Figs. 6 and 7.
2008 *Hemipristis serra*; Tao & Hu, pl. 6, Fig. 1.

**Referred specimens:** $n = 7$: ASIZF0100460–0100462, CMM F0232, F2826, F2827, NTM I01131_2.

**Description:** CH = 5.21–30.81 mm; MCL = 8.73–41.38 mm; BCW = 6.50–36.59 mm. All collected specimens of this taxon represent upper teeth that are characterized by a distally inclined, broad triangular crown, and a mesiodistally separated bilobate root. Coarse serrations are present along the distal cutting edge, whereas serrations along the mesial cutting edge are finer. The root has a prominent lingual protuberance with a deep

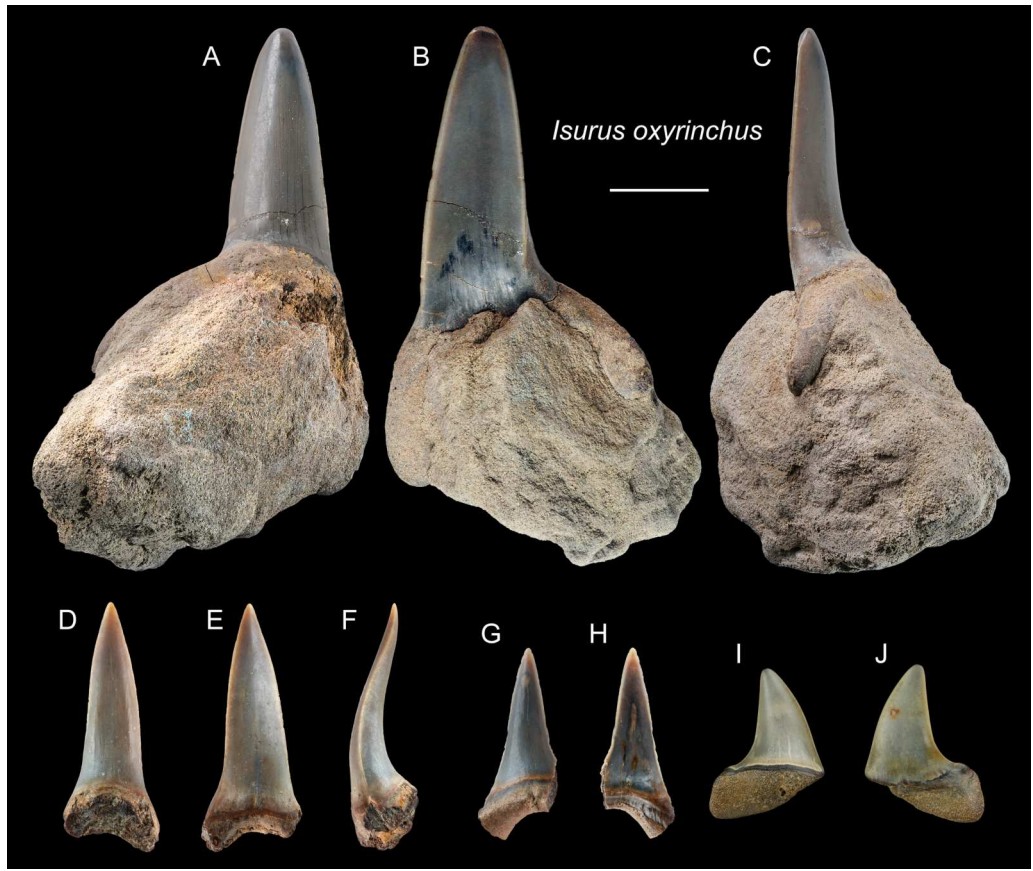

**Figure 5** **Teeth of *Isurus oxyrinchus* from the early Pleistocene, Liuchungchi Formation of Niubu, southern Taiwan.** (A–C) ASIZF0100317; (D–F) ASIZF0100318; (G and H) ASIZF0100321; (I and J) CMM F0242. (A, D, G and I) lingual views; (B, E, H and J) labial views; (C and F) lateral views. Scale bar = 1 cm.

nutritive groove, and has a notch-like shallow basal concavity. The crown overhangs the root, and the crown-root boundary, especially on the lingual face, is strongly arched.

   **Remarks:** As presumed sister species, the teeth of extinct †*Hemipristis serra* and extant *H. elongata* are similar. However, compared to †*H. serra*, teeth of *H. elongata* possess a more gracile crown, a longer apex without serration, and a narrower root (*Smith, 1957*; *Purdy et al., 2001*). The Pleistocene records of †*H. serra* are rare globally compared to its Neogene records (*Hooijer, 1954*; *Hooijer, 1958*; *Yabumoto & Uyeno, 1994*; *Carrillo-Briceño et al., 2015*; *Ebersole, Ebersole & Cicimurri, 2017*; *Boessenecker, Boessenecker & Geisler, 2018*).

   Family Carcharhinidae *Jordan & Evermann, 1896*
   Genus *Carcharhinus Blainville, 1816*

   **Remarks:** The identification based on teeth below the genus level is difficult for *Carcharhinus* (*Compagno, 1984*; *Compagno, 1988*; *Purdy et al., 2001*; *Naylor & Marcus,*

*1994*; *Marsili, 2006*; *Voigt & Weber, 2011*; *Ebert, Dando & Fowler, 2021*). Most of the upper teeth are triangular with crowns inclining distally. In different species, the crown varies from narrow to broad, and has smooth to coarsely serrated cutting edges, variable notch angles on distal cutting edges, and the straight to convex mesial cutting edge. At least nine species of *Carcharhinus* are recorded in the collections: *C. altimus*, *C. amboinensis*, *C. leucas*, *C. limbatus*, *C. longimanus*, *C. obscurus*, *C. plumbeus*, *C. sorrah*, and *C. tjutjot*. See remarks below for comparisons among other similar-looking species.

*Carcharhinus altimus* *Springer, 1950*
(Fig. 7)

**Referred specimens:** $n = 17$: ASIZF0100357, 0100359, 0100362, 0100363, 0100365, CMM F0080, F0101, F0113, F0134, F0214, F0224, F0293, F0304, F0322, F0363, TNM I01125, I01129_1.

**Description:** CH = 4.55–9.82 mm; MCL = 7.91–12.72 mm; BCW = 7.10–10.92 mm. The specimens examined in this study consist only of upper teeth. The crown of the upper teeth is finely serrated and varies in shape from a tall triangle to distally oblique. There is a notch on the distal cutting edge, whereas a slight constriction occurs on the lower part of the mesial cutting edge. The root is arched and has a nutritive groove. The roots of some specimens are not well-preserved (Figs. 7A, 7B, 7E, 7F, 7I–7L), but where well-preserved (Figs. 7C, 7D, 7G and 7H), it is arched and exhibits a nutritive groove on the lingual face.

**Remarks:** Teeth of *Carcharhinus altimus* and *C. plumbeus* are similar. However, those of *C. altimus* exhibit a distally bent apex unlike those of *C. plumbeus* that show a straight apex (Figs. 7 vs. 13).

*Carcharhinus amboinensis* *Müller & Henle, 1839*
(Fig. 8)

**Referred specimens:** $n = 5$: ASIZF0100366, 0100368, 0100369, CMM F0209, F0229.

**Description:** CH = 6.88–8.95 mm; MCL = 9.28–14.74 mm; BCW = 9.16–16.86 mm. The triangular crown is broad and exhibits coarse serrations although the serrations become smaller towards the apex. A prominent tooth neck is present between the crown and root on the lingual face. There is a notch on the distal cutting edge, whereas the mesial cutting edge is nearly straight. The bilobed root is gently arched and has a nutritive groove on the lingual face.

**Remarks:** Teeth of *Carcharhinus amboinensis*, *C. leucas*, and *C. longimanus* are very similar (*Marsili, 2006*; *Voigt & Weber, 2011*). However, the angle of the notch on the distal cutting edge of *C. longimanus* is larger than *C. leucas* and *C. amboinensis*. Compared to the teeth of *C. leucas*, the upper teeth of *C. amboinensis* are somewhat broader, the crowns are generally lower and more distally curved, and their distal heel is more pronounced and is closer to the base of the crown (*Kocsis et al., 2019*).

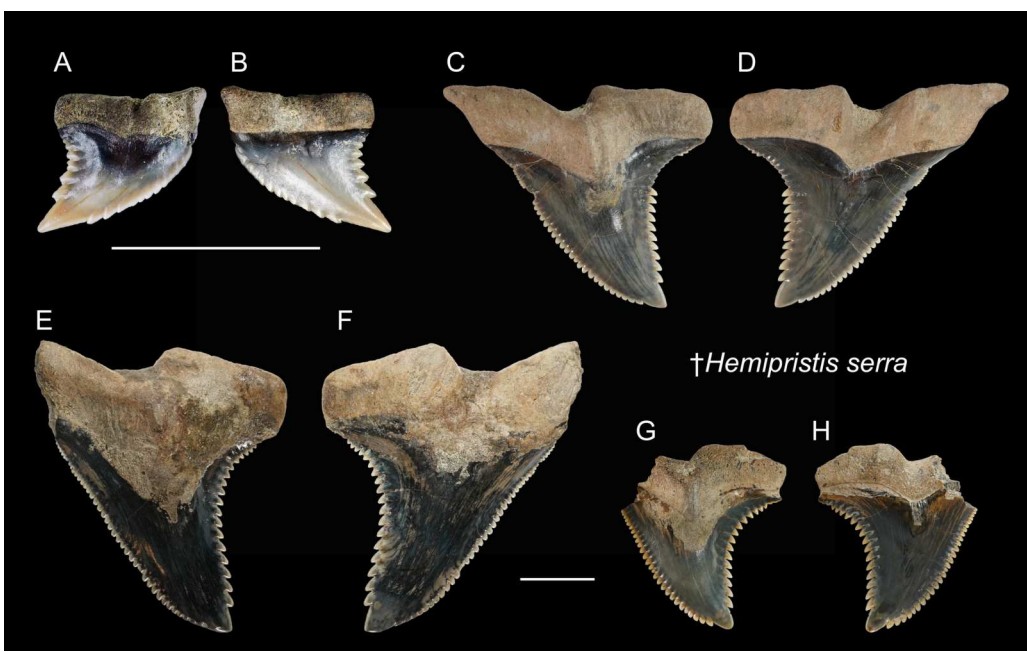

**Figure 6** Teeth of †*Hemipristis serra* from the early Pleistocene, Liuchungchi Formation of Niubu, southern Taiwan. (A and B) CMM F0232; (C and D) ASIZF0100460; (E and F) ASIZF0100461; (G and H) ASIZF0100462. (A, C, E and G) lingual views; (B, D, F and H) labial views. Scale bars = 1 cm.

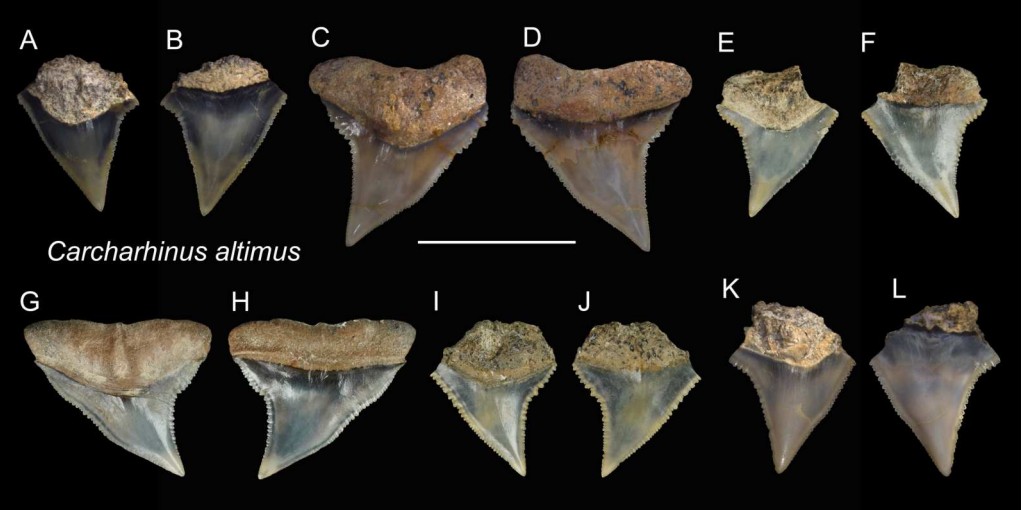

**Figure 7** Teeth of *Carcharhinus altimus* from the early Pleistocene, Liuchungchi Formation of Niubu, southern Taiwan. (A and B) ASIZF0100357; (C and D) ASIZF0100359; (E and F) CMM F0363; (G and H) CMM F0293; (I and J) CMM F0322; (K and L) ASIZF 0100365. (A, C, E, G, I and K) lingual views; (B, D, F, H, J and L) labial views. Scale bar = 1 cm.

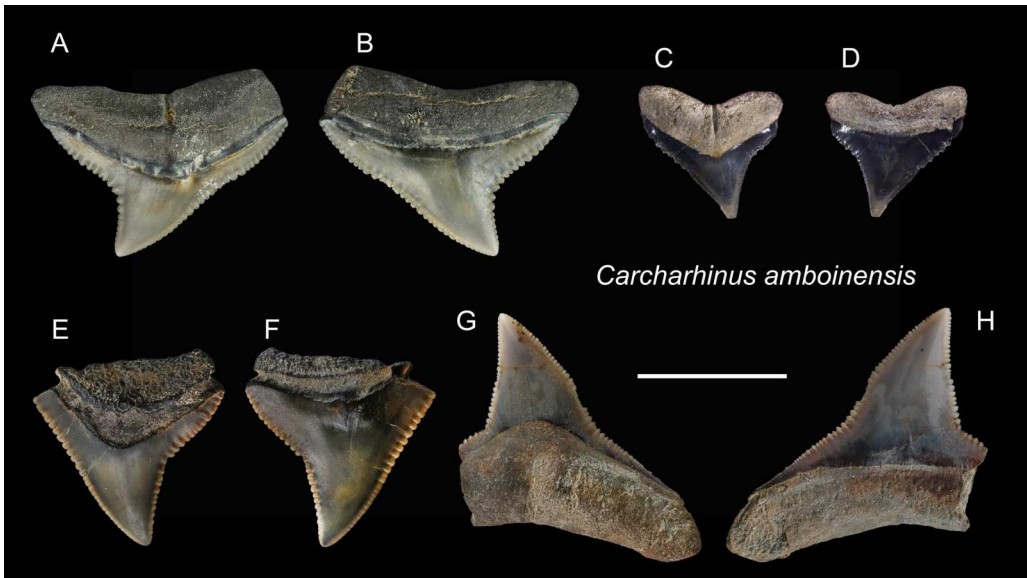

**Figure 8** Teeth of *Carcharhinus amboinensis* from the early Pleistocene, Liuchungchi Formation of Niubu, southern Taiwan. (A and B) CMM F0209; (C and D) ASIZF0100368; (E and F) ASIZF0100366; (G and H) ASIZF0100369. (A, C, E and G) lingual views; (B, D, F and H) labial views. Scale bar = 1 cm.

*Carcharhinus leucas* Valenciennes, 1839
(Fig. 9)
?1965 *Carcharhinus gangeticus*; Huang, pl. 22, Figs. 19 and 20.
2004 Elasmobranchii indet.; Xue, pl. 3, Fig. 2, pl. 4, Fig. 3, pl. 7, Fig. 7, pl. 9, Fig. 4.

**Referred specimens:** $n = 71$: ASIZF0100390, 0100393–0100398, 0100400–0100404, 0100411, 0100419, 0100424, 0100425, 0100481, CMM F0154, F0155, F0157, F0159, F0162, F0163, F0165–F0168, F0170–F0175, F0180, F0183, F0186–F0188, F0190, F0192, F0198–F0201, F0205, F0206, F0221, F0222, F0227, F0231, F0240, F0244, F0246, F0249, F0288, F0290, F0297, F0299, F0301, F0317, F0319, F0321, F0328, F0332, F0334, F0341, F0342, F0348, F0354, F0362, NTM I01130_2.

**Description:** CH = 4.87–18.68 mm; MCL = 7.97–21.69 mm; BCW = 8.73–30.56 mm. The teeth of *Carcharhinus leucas* are generally robust. The crown of the upper teeth (Figs. 9A–9P) is broad and triangular with a slight distal inclination. The middle of the distal cutting edge is concave, forming a weak notch, whereas the mesial cutting edge is straight to slightly convex. Both cutting edges are coarsely serrated, but the sizes of serrations are smaller at the base and apex of the crown than those in the middle. The boundary between the crown base and root on the lingual face displays a V-shape tooth neck. The bilobate root is arched and displays a weak nutritive groove on the lingual face (Figs. 9A–9H, 9K and 9L). The lower teeth (Figs. 9Q and 9R), that have fine serrations, are labiolingually thicker and mesiodistally narrower than the upper teeth.

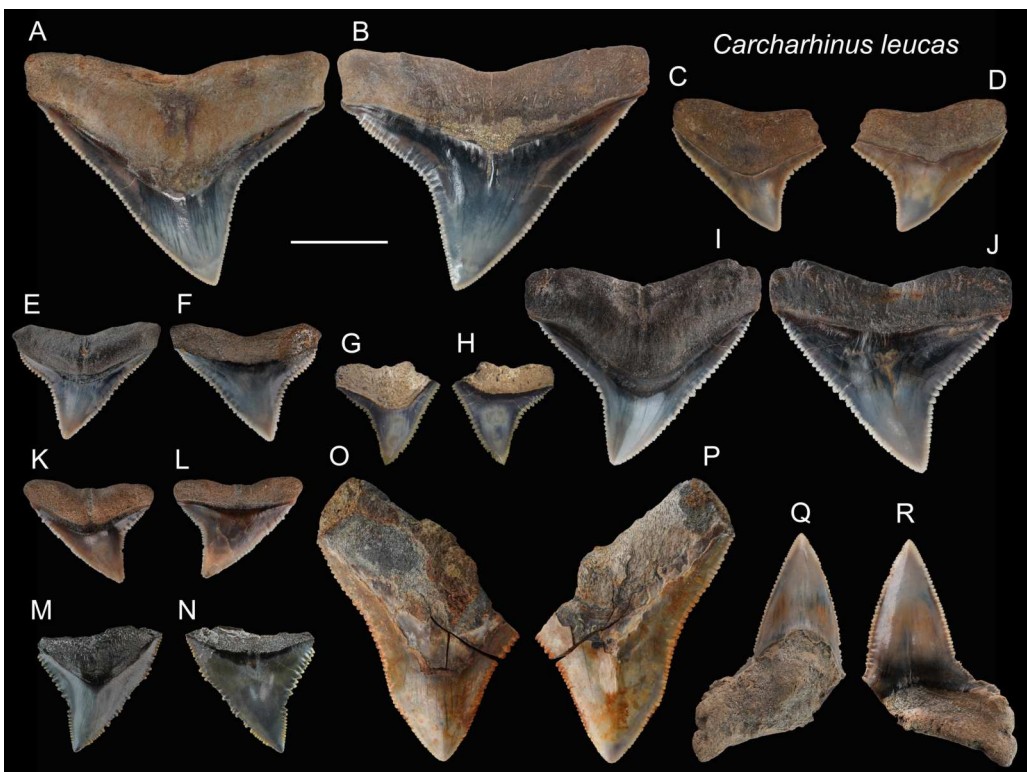

**Figure 9  Teeth of *Carcharhinus leucas* from the early Pleistocene, Liuchungchi Formation of Niubu, southern Taiwan.**  (A and B) ASIZF0100398; (C and D) ASIZF0100397; (E and F) ASIZF0100394; (G and H) ASIZF0100411; (I and J) ASIZF0100396; (K and L) ASIZF 0100395; (M and N) ASIZF0100400; (O and P) ASIZF0100402; (Q and R) ASIZF0100390. (A, C, E, G, I, K, M, O and Q) lingual views; (B, D, F, H, J, L, N, P and R) labial views. Scale bar = 1 cm.

**Remarks:** *Marsili (2006)* described the crown of *Carcharhinus longimanus* as larger, more elongate and possessing a straighter root margin compared to that of *C. leucas*. In addition, based on the images of *Carcharhinus* by *Garrick (1982)* and *Voigt & Weber (2011)*, we find some other slight differences in tooth morphology between the two species. For example, the angle on the distal cutting edge of the upper teeth in *C. longimanus* is larger than that in *C. leucas*, making the crown of *C. leucas* incline more distally than that in *C. longimanus*. In addition, the tooth shape of *C. leucas* is close to a wide-bottom triangle, whereas that of *C. longimanus* forms a taller triangle. Furthermore, the lower teeth of *C. leucas* tend to exhibit a stronger demarcation between the main cusp and mesial and distal heels than those of *C. longimanus* with a smoother cusp-heel transition.

*Carcharhinus limbatus Valenciennes, 1839*
(Fig. 10)
1978 *Carcharhinus* sp.; Uyeno, pl. 3, Fig. 14.
2004 Elasmobranchii indet.; Xue, pl. 8, Fig. 5.

Referred specimens: $n = 40$: ASIZF0100467–0100480, 0100482, 0100483, CMM F0056, F0111, F0216, F0217, F0234, F0236–F0238, F0286, F0289, F0291, F0295, F0306, F0307, F0310, F0368–F0373, NTM I01127, I01133_2, I01134_2.

Description: CH = 7.70–9.31 mm; MCL = 10.02–13.26 mm; BCW = 10.26–14.70 mm. Our specimens consist only of upper teeth. The teeth of *C. limbatus* are serrated and are characterized by a narrow cusp that is erect to slightly oblique distally with a mesiodistally wide crown base. The serrations near the crown base are coarser than those towards the apex. The root is apicobasally shallow. Its base is straight to slightly arched with a prominent deep nutritive groove that forms a notch along the root base.

Remarks: Although similar, teeth of *Carcharhinus limbatus* can be distinguished from those of *C. amblyrhychoides*, *C. brachyurus* and *C. brevipinna*. Unlike the teeth of *C. limbatus*, the serrations on the cutting edges tend not to continue to the crown base in *C. amblyrhychoides*, are absent or weak in *C. brevipinna*, and in *C. brachyurus*, the apex is more pointed and more distally directed than in *C. limbatus* (*Garrick, 1982*; *Voigt & Weber, 2011*). In addition, the crowns of *C. limbatus* have a narrow, erect cusp with a sharp transition to a broad crown base that is distinct from all other congeneric specimens examined. The teeth of *C. limbatus* and *C. amblyrhynchoides* are, however, very difficult to distinguish. *Kocsis et al. (2019)* noted a narrower crown with finer serrations in *C. limbatus*, but this character is not clear in our specimens. Currently, no records of *C. amblyrhynchoides* have been reported in Taiwan (*Ebert et al., 2013*; *Shao, 2022*); therefore, we tentatively assign these specimens to *C. limbatus*.

*Carcharhinus longimanus* *Poey, 1861*
(Fig. 11)
1965 *Carcharhinus gangeticus*; Huang, pl. 22, (Figs. 21 and 22).
2004 Elasmobranchii indet.; Xue, pl. 4, Fig. 4, pl.7, Fig. 6, pl. 9, Fig. 1.

Referred specimens: $n = 36$: ASIZF0100370, 0100371, 0100373–0100382, 0100391, 0100392, 0100421, 0100422, 0100428, 0100466, CMM F0006, F0011, F0087, F0151, F0153, F0156, F0158, F0182, F0189, F0194, F0195, F0197, F0223, F0248, F0287, F0294, NTM I01128, NTM I01130.

Description: CH = 10.23–15.93 mm; MCL = 13.51–22.08 mm; BCW = 13.20–21.69 mm. The crowns of the upper teeth (Figs. 11A–11P) are broad, triangular, and coarsely serrated. The distal cutting edge is weakly concave, whereas the mesial cutting edge is nearly straight. The crown base on the lingual side is deeply concave and is accompanied basally by a narrow tooth neck and a deep bilobate root with a shallow nutritive groove. The lower teeth (Figs. 11Q–11T) are thicker and narrower than the upper teeth, they also have fine serrations on the cutting edges. The boundary between the crown base and root on the lingual side is also deeply concave with a V-shaped tooth neck.

Remarks: See remarks under *Carcharhinus leucas*.
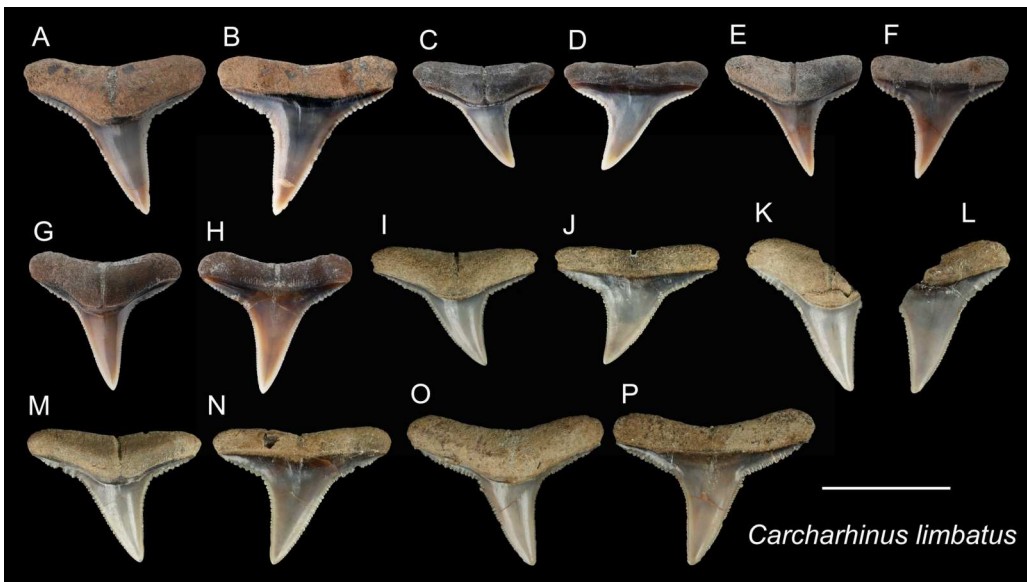

_Figure 10_ **Teeth of *Carcharhinus limbatus* from the early Pleistocene, Liuchungchi Formation of Ni-ubu, southern Taiwan.** (A and B) ASIZF0100470; (C and D) ASIZF0100476; (E and F) ASIZF0100469; (G and H) ASIZF0100468; (I and J) CMM F0236; (K and L) CMM F0111; (M and N) CMM F0237; (O and P) CMM F0238. (A, C, E, G, I, K, M), and O) lingual views; (B, D, F, H, J, L, N and P) labial views. Scale bar = 1 cm.

*Carcharhinus obscurus Lesueur, 1818*
(Fig. 12)
2004 Elasmobranchii indet.; Xue, pl. 4, Fig. 7.

**Referred specimens:** $n = 25$: ASIZF0100372, 0100383–0100389, 0100399, CMM F0123, F0143, F0148, F0160, F0164, F0176–F0179, F0181, F0184, F0196, F0208, F0338, F0353, NTM I1132_3.

**Description:** CH = 5.04–15.14 mm; MCL = 7.57–21.61 mm; BCW = 9.56–20.96 mm. The specimens in this study consist only of upper teeth. They are broad and triangular with coarse serrations, although the serrations tend to become finer apically. The mesial cutting edge is overall slightly convex with a marked distally directed apex. The distal cutting edge has a relatively deep notch, but the degree of the angle varies based on tooth position within the dentition. The crown base on the lingual side is moderately concave and is accompanied by a prominent tooth neck and a relatively robust bilobed root that has a shallow nutritive groove.

**Remarks:** The crown of *Carcharhinus obscurus* is mesiodistally broad and typically exhibits coarse serrations along the middle section of both cutting edges, a feature for separating the species from all other congeneric specimens in the present study.

**Referred specimens:** $n = 51$, ASIZF0100405–0100410, 0100412, 0100429, CMM F0074–F0077, F0079, F0086, F0088, F0091, F0096, F0100, F0106, F0115, F0124, F0144, F0146,

Lin et al. (2022), *PeerJ*, DOI 10.7717/peerj.14190    16/39

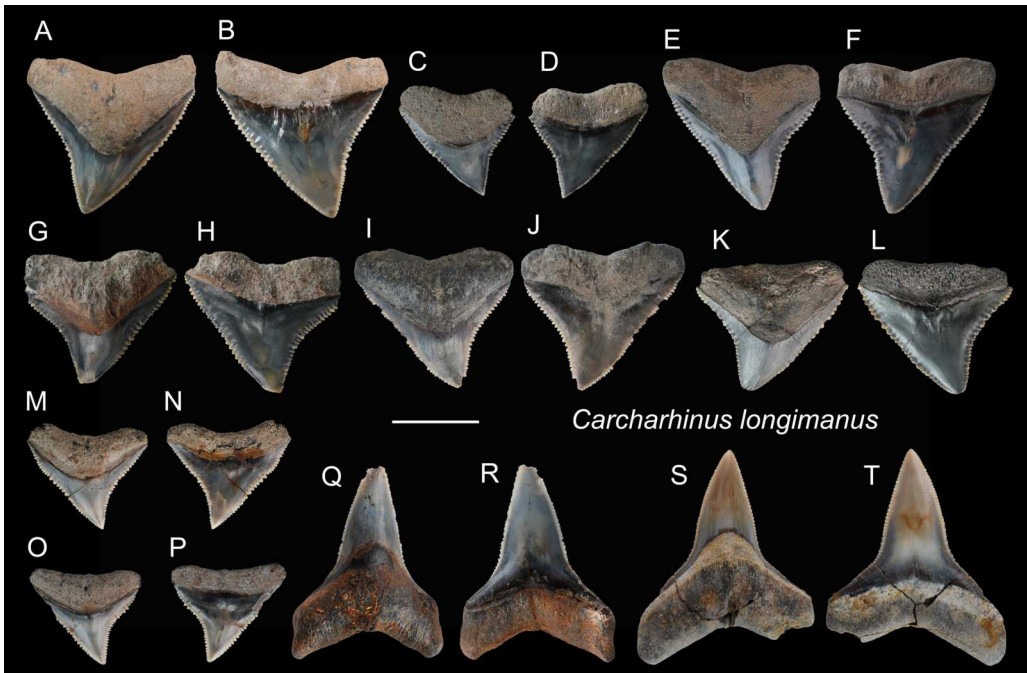

**Figure 11  Teeth of *Carcharhinus longimanus* from the early Pleistocene, Liuchungchi Formation of Niubu, southern Taiwan.** (A and B) ASIZF 0100371; (C and D) ASIZF0100376; (E and F) ASIZF0100370; (G and H) ASIZF0100377; (I and J) ASIZF0100375; (K and L) ASIZF0100378; (M and N) ASIZF0100374; (O and P) ASIZF0100373; (Q and R) ASIZF0100392; S, T, ASIZF0100391. (A, C, E, G, I, K, M, O, Q and S) lingual views; (B, D, F, H, J, L, N, P, R and T) labial views. Scale bar = 1 cm.

*Carcharhinus plumbeus Nardo, 1827*
(Fig. 13)

F0169, F0225, F0228, F0292, F0302, F0316, F0318, F0320, F0325–F0327, F0330, F0331, F0333, F0335, F0337, F0346, F0347, F0349, F0350, F0352, F0356, F0360, F0361, F0364, F0365, F0367, NTM I01124.

**Description:** CH = 6.28–12.17 mm; MCL = 7.81–17.06 mm; BCW = 7.48–13.39 mm. The teeth that are referred to this species are all upper teeth. They are triangular with a slight distal inclination and with fine serrations. The mesial cutting edge is nearly straight, whereas the distal cutting edge tends to form a shallow notch close to the crown base. The root is bilobate and arched, and a shallow nutritive groove is present on the lingual face.

**Remarks:** The crown of *Carcharhinus plumbeus* is narrower and more elongate than that of *C. leucas*, *C. longimanus*, *C. obscurus*, and *C. amboinensis*, but it is wider than that of *C. altimus*.

*Carcharhinus sorrah Valenciennes, 1839*
(Fig. 14)

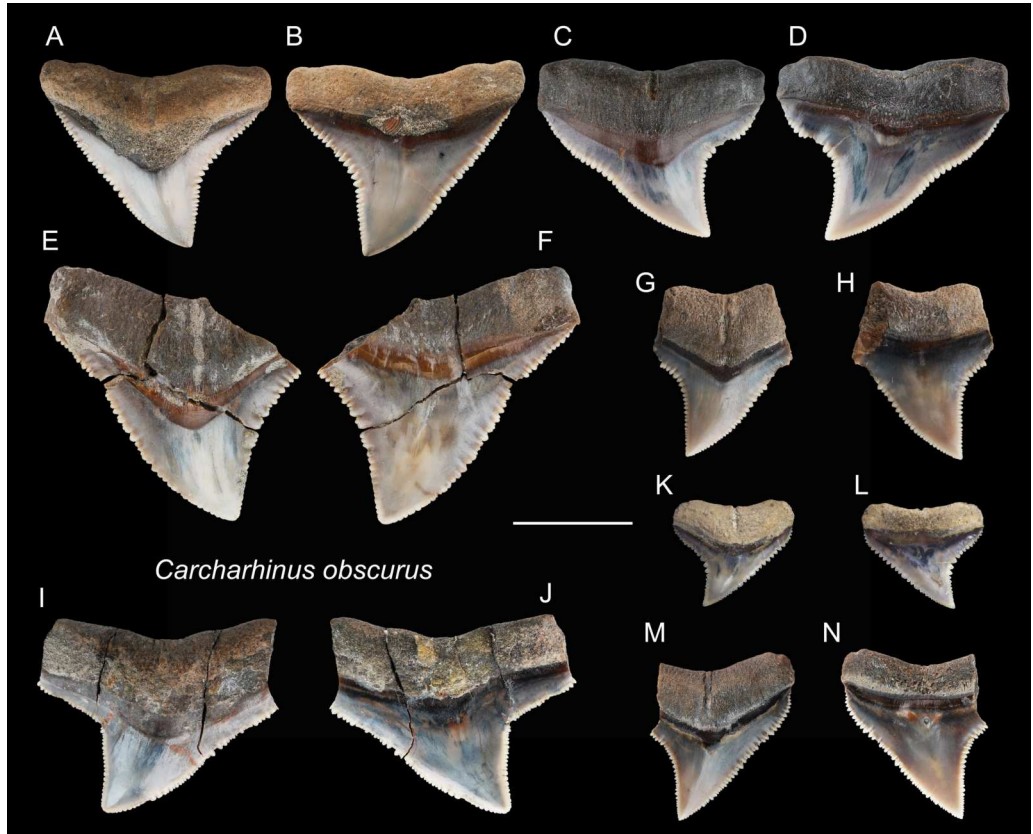

**Figure 12 Teeth of *Carcharhinus obscurus* from the early Pleistocene, Liuchungchi Formation of Niubu, southern Taiwan.** (A and B) ASIZF0100372; (C and D) ASIZF0100384; (E and F) ASIZF0100385; (G and H) ASIZF0100386; (I and J) ASIZF0100388; (K and L) ASIZF0100387; (M and N) ASIZF0100383. (A, C, E, G, I, K and M) lingual views; (B, D, F, H, J, L and N) labial views. Scale bar = 1 cm.

**Referred specimens:** $n = 11$: ASIZF0100418, CMM F0117, F0119, F0122, F0126, F0129, F0135, F0140, F0303, F0343, F0344.

**Description:** CH = 4.39–5.80 mm; MCL = 5.55–9.82 mm; BCW = 4.03–9.85 mm. All teeth identified to this species are represented by upper teeth. Their crowns exhibit finely serrated triangular cusps that strongly incline distally along with a coarsely serrated, relatively broad distal heel. The apex is narrow and may be slightly recurved (Figs. 14E–14H). The serrations on the distal heel become smaller distally, where finer secondary serrations are observed on one or two of the mesial-most serrations. Well-preserved specimens exhibit a strong nutritive groove on the lingual face that forms a notch along the root base.

**Remarks:** According to *Voigt & Weber (2011)*, the crown of the upper teeth in *Carcharhinus sorrah* is high, and its distal cutting edge is deeply notched. These features are seen in our specimens; however, the description of the serrations in *Voigt & Weber (2011)* differs. The serrations on the central part of the mesial cutting edges are coarser in *Voigt & Weber (2011)*, whereas in our specimens, the coarsest serrations are on the basal part of the

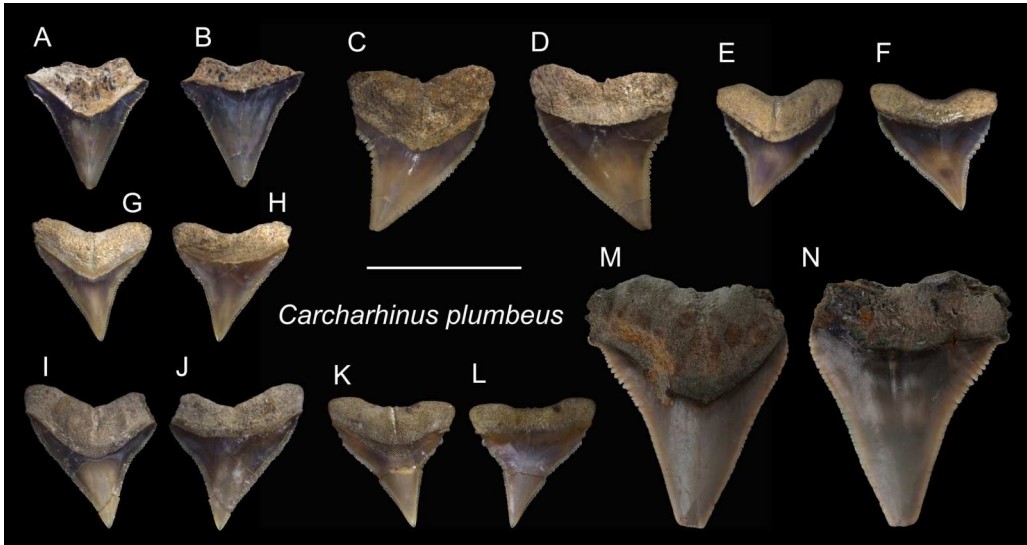

**Figure 13** **Teeth of *Carcharhinus plumbeus* from the early Pleistocene, Liuchungchi Formation of Niubu, southern Taiwan.** (A and B) ASIZF0100412; (C and D) ASIZF0100406; (E and F) ASIZF0100405; (G and H) ASIZF0100410; (I and J) ASIZF0100409; (K and L) ASIZF0100408; (M and N) ASIZF0100407. (A, C, E, G, I, K and M) lingual views; (B, D, F, H, J, L and N) labial views. Scale bar = 1 cm.

mesial cutting edges. Furthermore, the main cusp and coarse serrations in our specimens are farther apart than those figured by *Voigt & Weber (2011)*. The teeth of *C. tjutjot* and *C. sorrah* are both characterized by a coarsely serrated distal heel, but the teeth of *C. tjutjot* differ from those of *C. sorrah* by having fewer but larger serrations forming a distal heel (Figs. 14 vs. 15).

*Carcharhinus tjutjot Bleeker, 1852*
(Fig. 15)

**Referred specimens:** *n* = 19: ASIZF0100413–0100417, CMM F0116, F0136–F0138, F0142, F0296, F0298, F0323, F0324, F0339, F0345, F0357, F0376, F0377.

**Description:** CH = 4.25–5.82 mm; MCL = 6.03–9.01 mm; BCW = 5.61–7.94 mm. The specimens of this species described here are all represented by upper teeth. They have a robust, distally inclined, triangular cusp with a small distal heel consisting of coarse serrations that rapidly diminish in size distally. The strongly inclined mesial cutting edge is relatively straight, where the apex may slightly recurve and serrations become slightly coarser towards the base. Finer secondary serrations are observed on the first and possibly second mesial-most serrations on the distal heel. The root is weakly bilobate and the root base is nearly straight. Well-preserved specimens show a shallow nutritive groove on the lingual face of the root.

**Remarks:** The teeth of *Carcharhinus sealei*, *C. dussumieri*, *C. coatesi*, and *C. tjutjot* are very similar (*White, 2012*). The difference between species is related to their serrations.

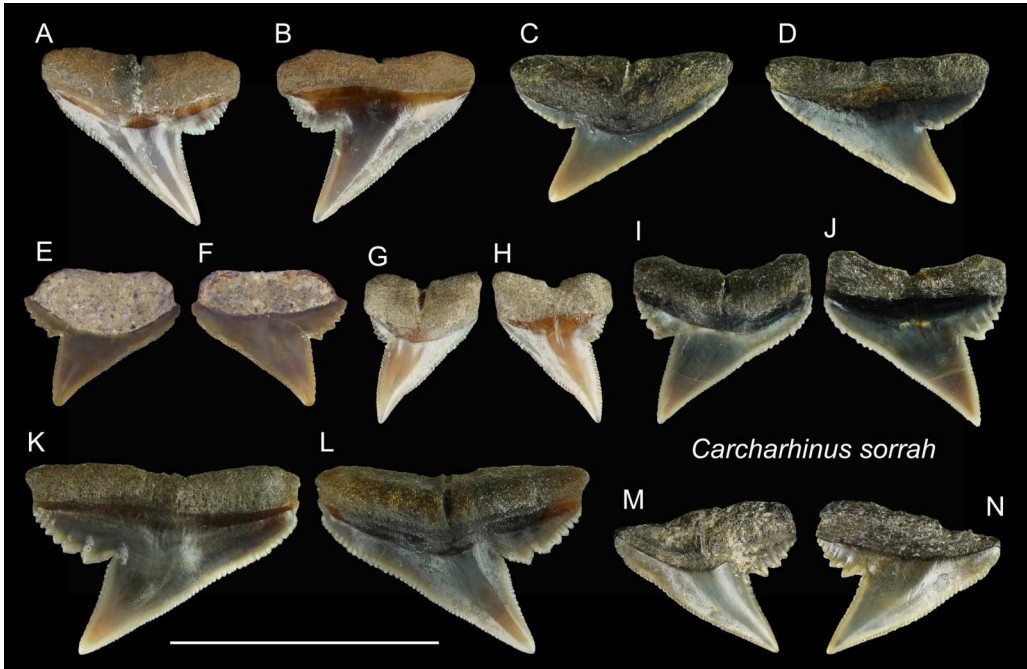

**Figure 14  Teeth of *Carcharhinus sorrah* from the early Pleistocene, Liuchungchi Formation of Niubu, southern Taiwan.** (A and B) CMM F0129; (C and D) CMM F0119; (E and F) ASIZF0100418; (G and H) CMM F0126; (I and J) CMM F0135; (K and L) CMM F0122; (M and N) CMM F0140. (A, C, E, G, I, K and M) lingual views; (B, D, F, H, J, L and N) labial views. Scale bar = 1 cm.

The serrations of *C. sealei* are present only on the basal half of the mesial cutting edge, whereas the distal cutting edge, including the distal heel, is smooth (*White, 2012*). Cutting edges of teeth in *C. coatesi* have fine to coarse serrations, but the distal heel is smooth. In *C. dussumieri*, both cutting edges, including the distal heel, have evenly-sized coarse serrations. The teeth of *C. tjutjot* also have evenly-sized serrated cutting edges, including the distal heel. *Carcharhinus dussumieri* and *C. tjutjot* have long been misidentified due to their similar appearance, but *C. dussumieri* is now considered a West Indian species distributed from the Persian Gulf to India, whereas *C. tjutjot* is distributed from Indonesia to Taiwan (*White, 2012*).

Genus *Rhizoprionodon Whitley, 1929*
*Rhizoprionodon acutus Rüppell, 1837*
(Fig. 16)

**Referred specimens:** *n* = 8: ASIZF0100463, 0100464, CMM F0110, F0120, F0121, F0130, F0131, F0218.

**Description:** CH = 3.97–5.35 mm; MCL = 6.22–10.82 mm; BCW = 7.68–10.69 mm. The upper teeth of this species have a crown that is strongly inclined distally and is accompanied by a low distal heel (Figs. 16A–16H). Both cutting edges, including the distal

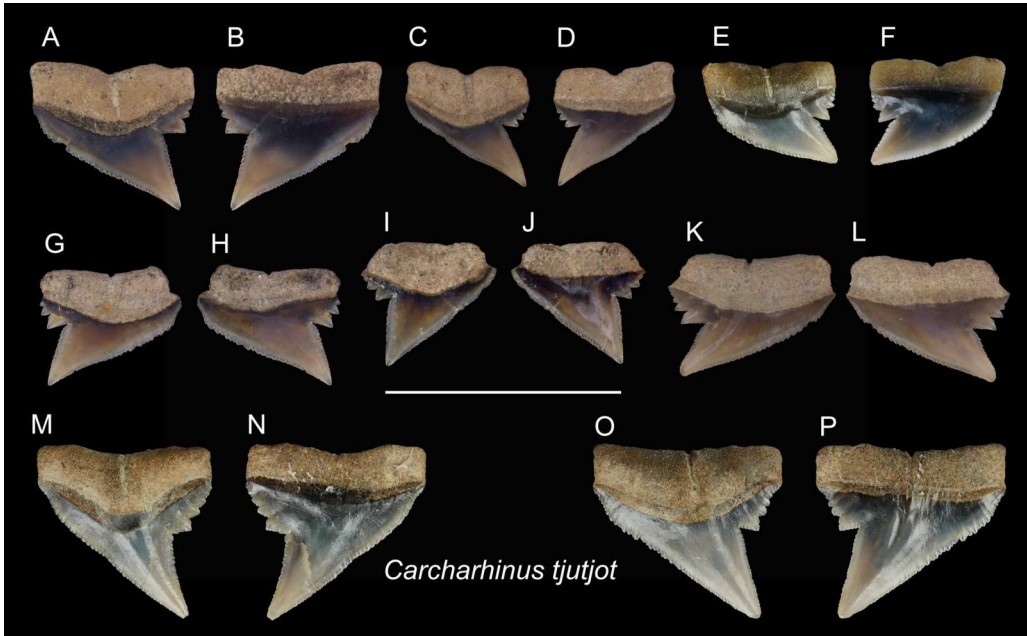

**Figure 15** Teeth of *Carcharhinus tjutjot* from the early Pleistocene, Liuchungchi Formation of Niubu, southern Taiwan. (A and B) ASIZF0100415; (C and D) ASIZF0100414; (E and F) CMM F0116; (G and H) ASIZF0100413; (I and J) ASIZF0100417; (K and L) ASIZF0100416; (M and N) CMM F0323; (O and P) CMM F0324. (A, C, E, G, I, K, M and O) lingual views; (B, D, F, H, J, L, N and P) labial views. Scale bar = 1 cm.

heel, are smooth or exhibit fine irregular serrations. The mesial cutting edge is overall straight, whereas the junction between the cusp and distal heel is deeply notched. A deep nutritive groove is present on the lingual side of the root that continues to the root base. The root is low with little to no basal concavity. ASIZF0100464 (Figs. 16I and 16J) is a lower tooth, with a crown that is unserrated and more gracile than the upper teeth with a concave mesial cutting edge. The root morphology is similar to that of lower teeth.

**Remarks:** The teeth of *Rhizoprionodon acutus* are serrated in adults (*Compagno, 1984*). In our specimens, the serrations are absent, indicating immature individuals. Distinguishing between the teeth of *R. acutus* and *R. oligolinx* is difficult, where both have very fine irregular serrations. However, due to the questionable distribution of *R. oligolinxi* in Taiwan (*Ebert et al., 2013*; *Froese & Pauly, 2022*), we tentatively assign these specimens to *R. acutus*.

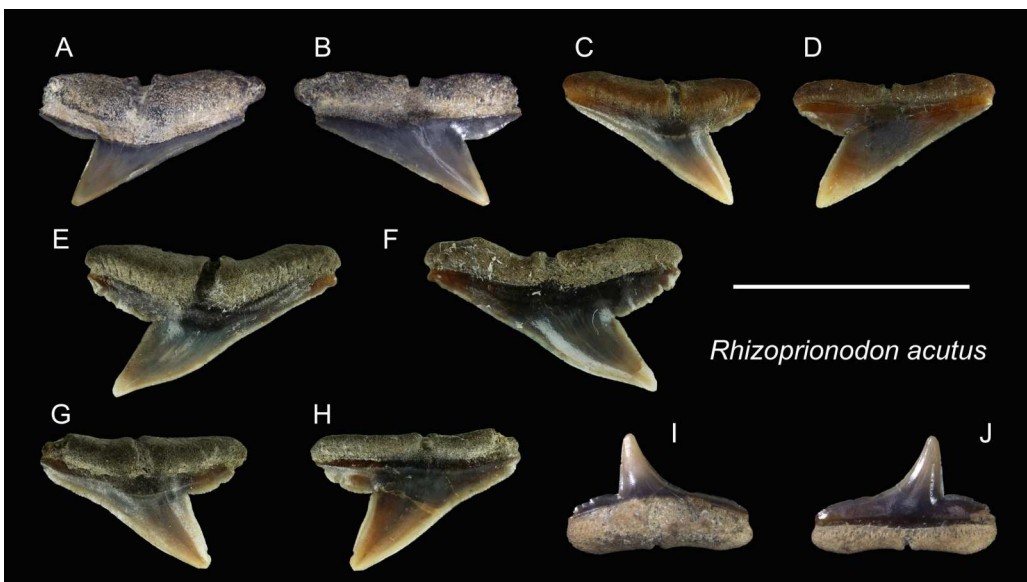

**Figure 16** **Teeth of *Rhizoprionodon acutus* from the early Pleistocene, Liuchungchi Formation of Niubu, southern Taiwan.** (A and B) ASIZF0100463; (C and D) CMM F0120; (E and F) CMM F0121; (G and H) CMM F0131; (I and J) ASIZF0100464. (A, C, E, G and I) lingual views; (B, D, F, H and J) labial views. Scale bar = 1 cm.

Family Galeocerdonidae *sensu Ebersole, Cicimurri & Stringer, 2019*
Genus *Galeocerdo Müller & Henle, 1837*
*Galeocerdo cuvier Péron & Lesueur, 1822*
(Figs. 17A–17H)
?1965 *Galeocerdo aduncus*; Huang, pl. 22, Figs. 10 and 11.
?1978 *Galeocerdo aduncus*; Uyeno, pl. 1, Fig. 3.
2004 Elasmobranchii indet.; Xue, pl. 6, Fig. 1–7, pl. 8, Fig. 6.

**Referred specimens:** *n* = 7: ASIZF0100459, CMM F0213, F0215, F0245, F2823, F2829, NTM I01121.

**Description:** CH = 12.27–17.72 mm; MCL = 17.37–26.88 mm; BCW = 18.10–28.05 mm. The teeth of *Galeocerdo cuvier* are characterized by a coarsely serrated crown with a cusp that strongly curves distally and a prominent distal heel demarcated by a deep notch with an approximately 90 degrees angle along the distal cutting edge. Fine secondary serrations are present on the coarser primary serrations (Figs. 17E and 17H). The serrations on the distal heel in ASIZF0100459 (Figs. 17F and 17G) are weak and the width to crown height ratio suggests this tooth represents a posterior position. CMM F0245 and CMM F0215 are anterior teeth with well-marked serrations (Figs. 17A–17D).

**Remarks:** Five extinct species and one extant species of *Galeocerdo* are considered valid: the Eocene †*G. clarkensis* and †*G. eaglesomi*, Oligocene–late Miocene †*G. aduncus*, Miocene †*G. mayumbensis*, Pliocene †*G. capellini*, and the Pleistocene–Recent *G. cuvier* (*Purdy et al., 2001*; *Türtscher et al., 2021*). The specimens described here are identified as *G. cuvier*,

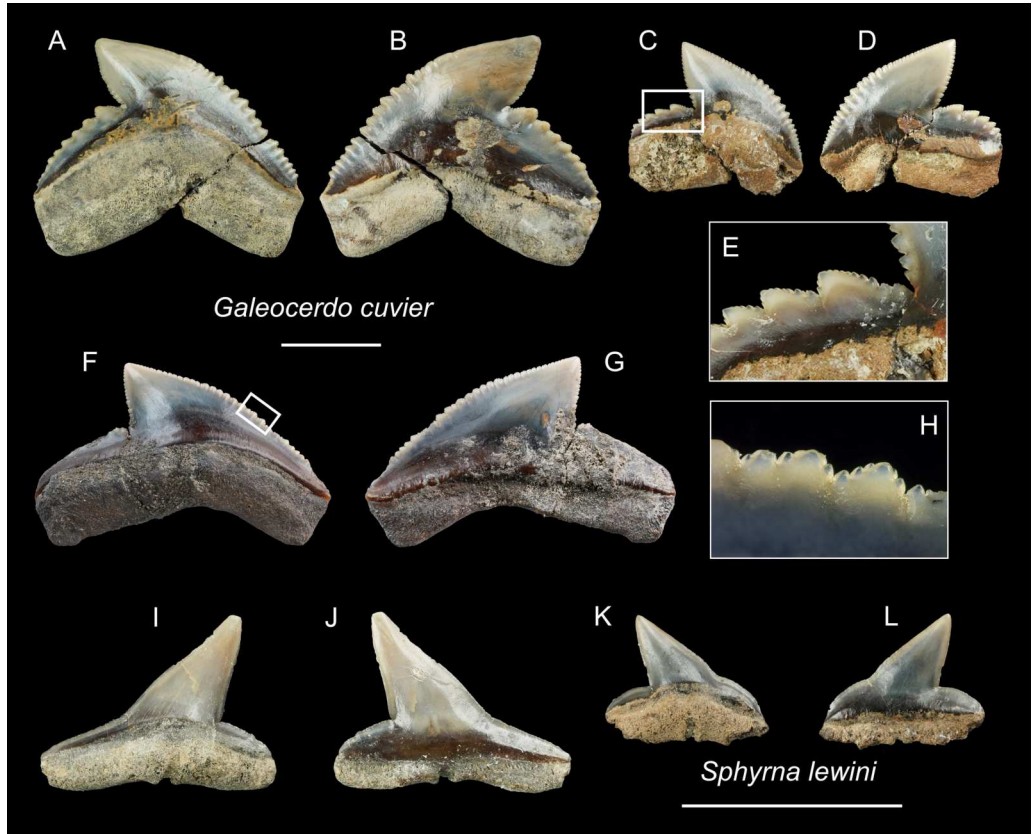

**Figure 17** **Teeth of *Galeocerdo cuvier* and *Sphyrna lewini* from the early Pleistocene, Liuchungchi Formation of Niubu, southern Taiwan.** (A–H) *Galeocerdo cuvier*; (A and B) CMM F0245; (C–E) CMM F0215; (F–H) ASIZF0100459. (I–L) *Sphyrna lewini*; (I and J) CMM F0235; (K and L) CMM F0312. (A, C, F, I and K) lingual views; (B, D, G, J and L) labial views; (E and H) details of secondary serrations. Scale bars = 1 cm.

particularly because of the presence of secondary serrations (*Cigala-Fulgosi & Mori, 1979*; *Türtscher et al., 2021*). *Huang (1965)* reported a questionable occurrence of †*G. aduncus* from the Pleistocene Cholan Formation in Hsinchu, northern Taiwan, but we consider the specimen lost. *Uyeno (1978)* reported another occurrence of †*G. aduncus* from the poorly constrained Plio-Pleistocene strata along the Tsailiao River in Tainan, southwestern Taiwan (as Miocene to Pleistocene in *Uyeno, 1978*). Although Uyeno's collection was deposited in the NTM, we were not able to locate the specimen of †*G. aduncus* in the collection. Nevertheless, although the whereabouts of the specimen is uncertain, it is interpreted here to have also belonged to *G. cuvier*.

Family Sphyrnidae *Bonaparte, 1840*
Genus *Sphyrna Rafinesque, 1810*
*Sphyrna lewini Griffith & Smith, 1834*
(Figs. 17I–17L)
?1978 *Sphyrna* sp.; Uyeno, pl. 2, Fig. 8

**Referred specimens:** $n = 2$: CMM F0235, F0312.

**Description:** CH = 4.64–6.85 mm; MCL = 7.51–11.17 mm; BCW = 7.19–10.53 mm. The tooth crown of *Sphyrna lewini* is characterized by a slender, distally inclined cusp with a narrow, mesially extended base separated by a slight concavity along the mesial cutting edge and a low distal heel demarcated by a deep notch. Both cutting edges are smooth without serrations. The root is low and its base is straight. It has a deep nutritive groove on the lingual side and extends to the root base.

**Remarks:** The teeth of *Sphyrna lewini* are most similar to *S. macrorhynchos* and *Loxodon macrorhinus*, but a slight concavity is present on the base of the mesial cutting edge in *S. lewini*, whereas the edge is almost straight in the latter two species (*Ebert et al., 2013*).

Order Myliobatiformes *Compagno, 1973*
Family Dasyatidae *Jordan & Gilbert, 1879*
Dasyatidae indet.
(Fig. 18)

**Referred specimens:** $n = 2$: ASIZF0100590, 0100591.

**Description:** The specimens are roughly hexagonal with a globular, thick crown and a well-divided bilobed root that is smaller than the crown and extends ventrally. The crown in both specimens is flat, but the specimen ASIZF0100590 (Figs. 18A–18D) has blunt, rounded corners compared to ASIZF0100591 (Figs. 18E–18H).

**Remarks:** The teeth referred to this taxon may belong to the genus *Dasystis* or *Himantura*, but because teeth of dasyatid taxa are highly variable in morphology, including sexual dimorphism and differences in ornamentation pattern (*Taniuchi & Shimizu, 1993*; *Kajiura & Tricas, 1996*; *Herman, Hovestadt-Euler & Hovestadt, 1998*; *Herman, Hovestadt-Euler & Hovestadt, 1999*; *Herman, Hovestadt-Euler & Hovestadt, 2000*), we refer our material simply to Dasyatidae indet. *Uyeno (1978)* reported teeth of *Dasyatis* sp. from the Miocene to Pleistocene of Taiwan. However, whether our specimens are conspecific with *Uyeno*'s (*1978*) specimens cannot be ascertained.

Family Aetobatidae *Agassiz, 1858*
Genus *Aetobatus Blainville, 1816*
*Aetobatus* sp.
(Fig. 19)

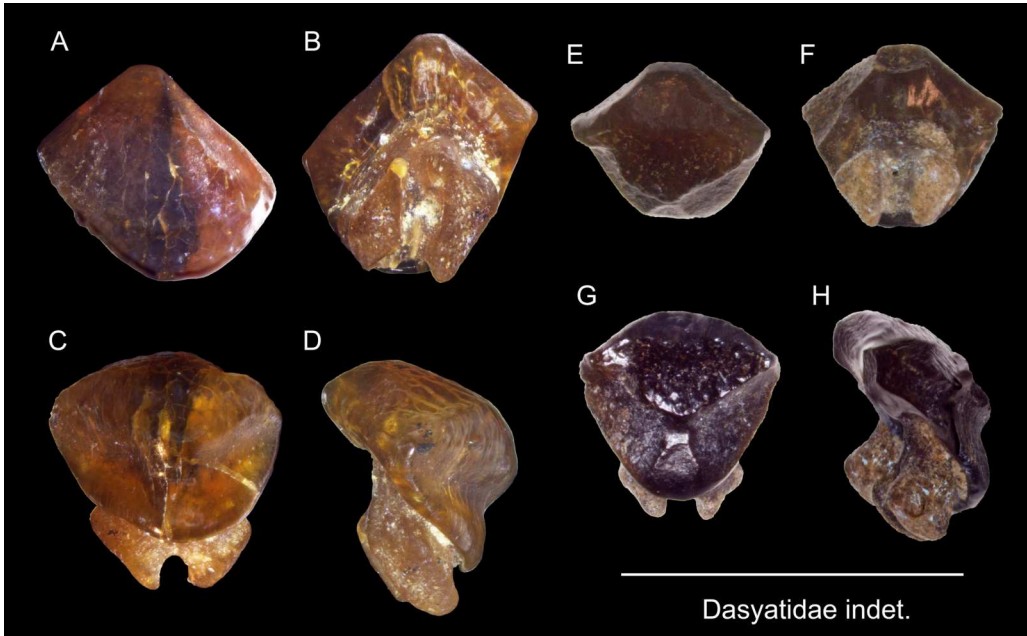

**Figure 18  Teeth of Dasyatidae indet. from the early Pleistocene, Liuchungchi Formation of Niubu, southern Taiwan.** (A–D) ASIZF0100590; (E–H) ASIZF0100591. (A and E) labial views; (B and F) basal views; (C and G) occlusal views; (D and H) lateral views. Scale bar = 5 mm.

**Referred specimens:** *n* = 58: ASIZF0100549–0100580, CMM F0380, F0382, F0388, F0395, F0399–0409, F0412, F2848–F2850, F2852–F2854, NTM I01116, I01117, I01119, I01120.

**Description:** Teeth of *Aetobatus* are characterized by strongly extended roots on the lingual (posterior) side and the arcuate crown in apical view with a flat occlusal surface. The crown overhangs the root on the labial (anterior) side and the root is more prominent than the crown on the lingual side. Both lingual and labial crown faces have fine vertical grooves as ornamentation. The root is polyaulocorhizous, consisting of anteroposteriorly oriented, densely packed, vertical lamellar plates.

**Remarks:** Five species of Myliobatidae (one *Aetobatus*, three *Aetomylaeus*, and one *Myliobatis*) are known from Taiwan (*Ebert et al., 2013*). All of which have grinding-type dental plates but each with different shapes and forms. The upper medial teeth of *Aetobatus ocellatus* are straight and elongate but slightly distally deflected towards the lingual side; its lower teeth are strongly arched towards the labial side. Considerable ontogenetic morphological change in dental plates is known in *Aetomylaeus* (*Hovestadt & Hovestadt-Euler, 2013*). Both upper and lower dental plates of adult *Aetomylaeus* are similar to the upper teeth of *Aetobatus*. Unlike adult individuals that have a single row of medial teeth, juveniles of *Aetomylaeus* have one medial, two lateral, and one posterior tooth row (*Hovestadt & Hovestadt-Euler, 2013*). The hexagon shape of medial teeth is very similar to those of juvenile *Myliobatis* (*Hovestadt & Hovestadt-Euler, 2013*). Teeth of *Aetobatus* have weak ornaments on the labial and lingual crown, but in *Aetomylaeus*, beaded ridges

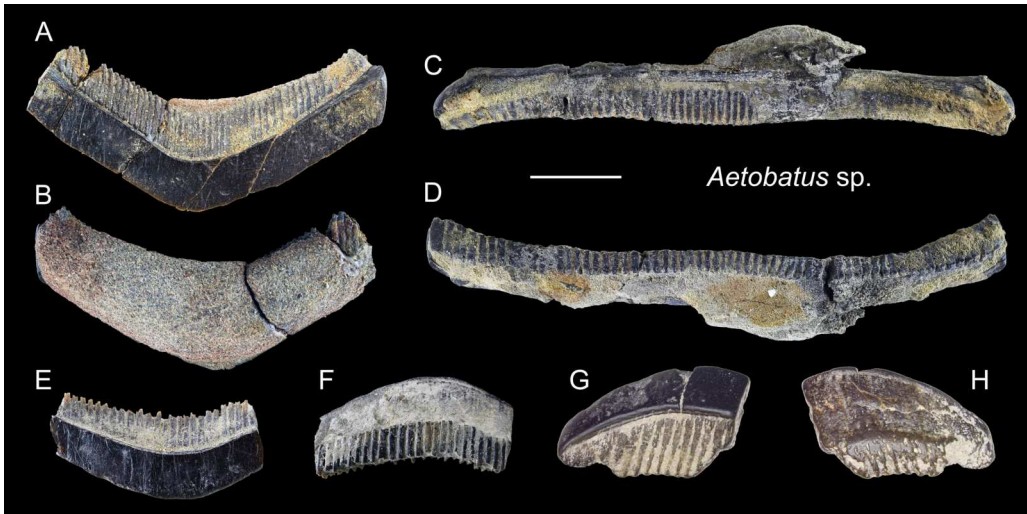

**Figure 19 Teeth of *Aetobatus* sp. from the early Pleistocene, Liuchungchi Formation of Niubu, southern Taiwan.** (A and B) CMM F2854; (C and D) CMM F2850; (E and F) CMM F0408; (G and H) ASIZF0100549. (A, C, E and G) occlusal views; (B, D, F and H) basal views. Scale bar = 1 cm.

with reticulated and pitting patterns are observed (*Ebersole, Cicimurri & Stringer, 2019*). Moreover, *Aetobatus* has teeth with a strong arched appearance than other myliobatid genera (see also remarks under *Myliobatis* sp.).

Family Myliobatidae *Bonaparte, 1835*
Genus *Myliobatis Cuvier, 1816*
*Myliobatis* sp.
(Fig. 20)

**Referred specimens:** *n* = 30: ASIZF0100581–0100589, CMM F0378, F0379, F0381, F0383–F0387, F0389–F0393, F0394, F0396–F0398, F0410, F0411, F2855, NTM I01118.

**Description:** Each tooth of *Myliobatis* has a flat occlusal surface and is laterally elongated and hexagonal that may be straight or slightly arched. The root is polyaulocorhizous with well-defined anteroposteriorly oriented, vertical lamellar plates separated by deep grooves, where the crown overhangs the root on the labial (anterior) face. The lingual and labial faces are ornamented with a network of fine reticulated ridges that grade into longitudinal ridges in the apical and become finer and anastomotic.

**Remarks:** The tooth plates of *Myliobatis* are similar to those of *Aetomylaeus* and *Aetobatus*, but the lateral angle of the hexagonal tooth plates in *Aetomylaeus* is more oblique than that of *Myliobatis* (*Ebersole, Cicimurri & Stringer, 2019*). The vertical lamellar plates of the root in *Myliobatis* are coarser than *Aetobatus*. Teeth of *Myliobatis* lack the tuberculated enameloid on the occlusal surface, whereas teeth of *Aetomylaeus* are reticulated on the labial and lingual faces (*Ebersole, Cicimurri & Stringer, 2019*). Because the total morphological variation range of teeth in many of the aetobatid and myliobatid (Myliobatinae) species

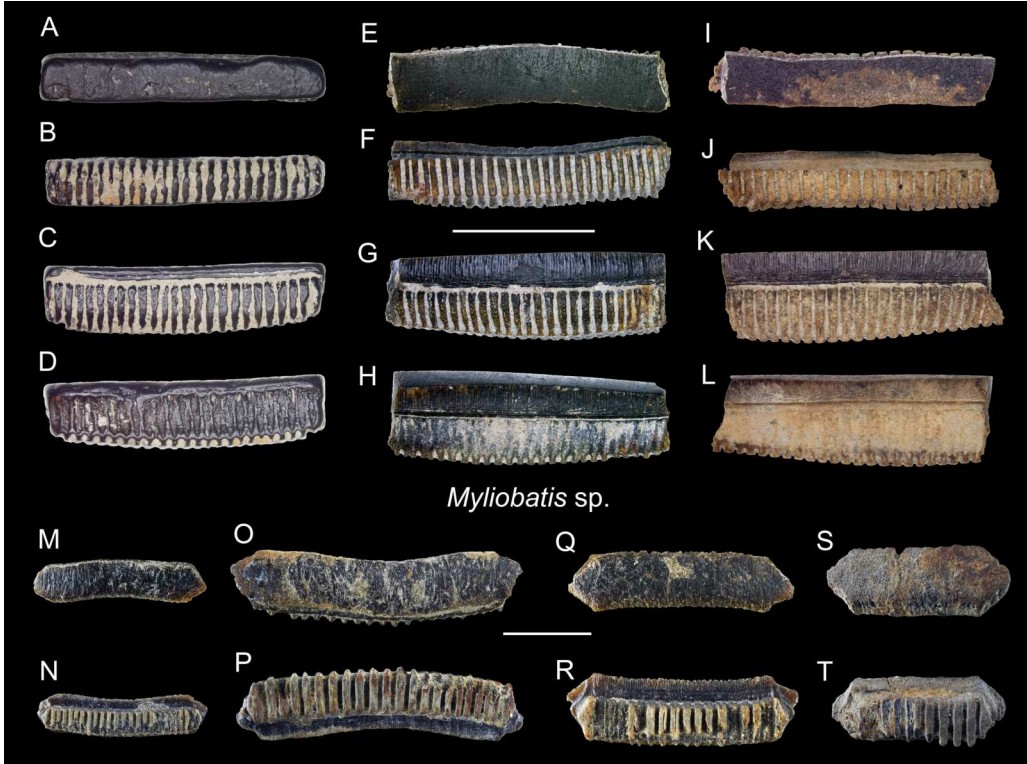

**Figure 20 Teeth of *Myliobatis* sp. from the early Pleistocene, Liuchungchi Formation of Niubu, southern Taiwan.** (A–D) ASIZF0100582; (E–H) ASIZF0100587; (I–L) ASIZF0100586; (M and N), CMM F0395; (O and P) CMM F2855; (Q and R) CMM F0393; (S and T) CMM F0398. (A, E, I, M, O, Q and S) occlusal views; (B, F, J, N, P, R and T) basal views; (C, G and K) lingual views; (D, H and L) labial views. Scale bars = 1 cm.

is unknown (*e.g.*, see *Hovestadt & Hovestadt-Euler, 2013*), we refrain from assigning the *Aetobatus* (see above) and *Myliobatis* teeth described here to the species level.

## DISCUSSION

Published work on fossil elasmobranchs in Taiwan is very scarce, limited in scope, often lacked formal descriptions, and were mostly based on private collections (*Lin et al., 2021*). *Huang (1965)* reported three shark taxa while describing a fossil whale tympanic bone from the early Pleistocene Cholan Formation in northern Taiwan (as early Pliocene in *Huang, 1965*). Although the whereabouts of the specimens is unknown, it is one of the earliest accounts reporting fossil shark teeth in Taiwan. *Uyeno (1978)* listed nine elasmobranch taxa from the Pleistocene Chochen–Tsailiao area with images of the specimens but without descriptions. These specimens are reviewed in the present study.

Perhaps the most complete description on a single fossil shark assemblage in Taiwan is the one by *Tao & Hu (2008)* from the late Miocene Tangenshan Sandstone in Chiahsien County, Kaohsiung. They described five taxa common in late Miocene marine deposits (*Otodus megalodon*, *Odontaspis* [*Carcharias?*] sp., †*Isurus hastalis*, †*Hemipristis serra*, and

**Table 2 Various diversity indices from the Pleistocene West Pacific elasmobranch assemblages showing high diversity of the present material.** See Table S1 for details of the data.

| Location | Age | Species richness | Shannon | Simpson | Fisher's alpha | Reference |
|---|---|---|---|---|---|---|
| Taiwan | early Pleistocene | 20 | 2.4 | 0.9 | 3.9 | Our study |
| Sulawesi | Pleistocene | 6 | 1.6 | 0.8 | 2.0 | *Hooijer (1954)* |
| Java | Pleistocene | 4 | 0.9 | 0.5 | 1.4 | *Koumans (1949)* |
| Java | Plio-Pleistocene | 11 | 1.7 | 0.7 | 2.9 | *Yudha et al. (2018)* |
| Central Japan | early Pleistocene | 2 | 0.5 | 0.3 | 1.2 | *Karasawa (1989)* |
| Central Japan | middle Pleistocene | 14 | 2.2 | 0.9 | 6.0 | *Kawase & Nishimatsu (2016)* |
| Eastern Japan | Pleistocene | 14 | 2.3 | 0.9 | 4.6 | *Tanaka & Taru (2022)* |

*Carcharhinus* sp.) as well as a new extinct species of *Hemipristis*, *H. liui* (*Tao & Hu, 2008*). †*Isurus hastalis* is now considered as Carcharodon hastalis (*Ehret et al., 2012*). The specimen of *H. liui* is an upper tooth and is characterized by asymmetric serrations on the distal and mesial cutting edges. The occurrences of *Otodus megalodon* are sparsely recorded from Taiwan (*Hu & Tao, 1993*; *Tao & Hu, 2008*) and are mostly present in private collections, which is a potential direction for future research efforts (*Haug et al., 2020*; *Lin et al., 2021*).

The materials reviewed in this study were mainly based on surface collecting that spanned over three decades. We note that our bulk sediment sampling (830 kg, see methods) only yielded three specimens (ASIZF0100548, ASIZF0100590, and ASIZF0100591). Surface collecting likely results in sampling bias towards larger specimens, underrepresenting smaller specimens (*Welton & Farish, 1993*; *Perez, 2022*). Nevertheless, 697 elasmobranch teeth from the Liuchungchi Formation in Niubu described in this study document the presence of at least 20 elasmobranch taxa (Table 1). The excellent overall preservation allowed species-level taxonomic identification for most of the specimens, which in turn, permitted the elucidation of the diverse elasmobranch community. In fact, the assemblage represents the most diverse elasmobranch paleofaunas from Taiwan reported to date.

The species richness and diversity indices suggest that our assemblage is highly diverse even with respect to other contemporaneous assemblages from temperate and tropical West Pacific (Table 2). Importantly, the number of specimens reported from other assemblages is much lower compared to our material (Table S1). However, the high species diversity in our collection likely reflects the geographic location of the study region, where both temperate and tropical species overlap and accumulate. Similar conditions are well recognized with marine fish faunas in Taiwan today (*Ebert et al., 2013*). Our present material indicates that this high diversity has preceded at least since the Pleistocene for the first time. Together, the high diversity captured in our study is significant in the spatio-temporal context.

The abundant and large teeth of *Carcharodon carcharias* are remarkable. *Carcharodon carcharias* is distributed along southern, eastern, and northeastern Taiwan today, but not on the west coast where the fossils are found (*Teng, 1958*; *Shen, 1993*; *Ebert et al., 2013*; *Shao, 2022*). According to the Fisheries Agency, Council of Agriculture, Taiwan (*Taiwan Fisheries Agency, 2021*), a total of 39 individuals of *C. carcharias* were caught between 2012 and 2021, with the majority of landings being in northeastern Taiwan. However, at our fossil sites, teeth of *C. carcharias* are the second most abundant fossils (*n* = 55) identified to the species

level in this study behind teeth of *Carcharhinus leucas* ($n = 71$, Table 1). Of the 55 isolated teeth that are interpreted to have most certainly come from 55 different individuals, 44 of them are well-preserved, allowing for tooth position identifications and accurate crown height measurements (CH). Based on the linear regression equation between the CH and total length (TL) for each tooth position in extant *Carcharodon carcharias* presented by *Shimada (2003)*, the CH of each of the 44 teeth was used to estimate the TL of each fossil individual (Table S2). Our specimens are normally distributed (Shapiro–Wilk test = 0.853, $p = 0.08$) and range in TL from 1.9 to 5.6 m, with a mean of 3.5 m (Table S2), suggesting the presence of many mature, large individuals (*Ebert et al., 2013*).

One of the most noteworthy occurrences reported in this study is that of the extinct species †*Hemipristis serra*. The species is known worldwide, but most of the documented occurrences are from the Miocene and Pliocene deposits (*e.g.*, *Yabumoto & Uyeno, 1994*; *Sánchez-Villagra et al., 2000*; *Marsili et al., 2007*; *Portell et al., 2008*; *Visaggi & Godfrey, 2010*; *Carrillo-Briceño et al., 2015*; *Kocsis et al., 2019*). The fossil record indicates that the fossil species preferred warm neritic environments (*Cappetta, 2004*; *Cappetta, 2012*). Although most previous studies suggest its last appearance at the end of the Pliocene, new evidence indicates that †*H. serra* persisted into the Pleistocene in North America (*Ebersole, Ebersole & Cicimurri, 2017*; *Boessenecker, Boessenecker & Geisler, 2018*; *Perez, 2022*). Teeth of *Hemipristis* that may belong to *H. serra* have been reported from Pleistocene and 'Plio-Pleistocene' deposits in Sulawesi and Java, Indonesia (*H.* cf. *serra* by *Hooijer, 1954*; *Hooijer, 1958*; simply "*Hemipristis*" by *Yudha et al., 2018*). Previous records of †*H. serra* from Taiwan were reported by *Uyeno (1978)* from an uncertain stratigraphic horizon along Tsailiao River, and that by *Tao & Hu (2008)* from the Miocene Kueichulin Formation in southern Taiwan. The †*H. serra* specimens described here are the first confirmed Pleistocene record in Taiwan, and along with the putative Indonesian records (*Hooijer, 1954*; *Hooijer, 1958*; *Yudha et al., 2018*), the geologically youngest records of the extinct species in the Northwest Pacific, meaning that the North American Pleistocene occurrences were not isolated.

The assemblage described here is dominated by two genera, *Carcharhinus* (Carcharhinidae, $n = 483$) and *Carcharodon* (Lamnidae, $n = 55$), which comprise more than 77.1% of the total specimen count and about half of the taxa identified (11 out of 20). From a paleoecological perspective, the composition is roughly similar to that found in modern western Taiwan (*Ebert et al., 2013*; *Shao, 2022*). For example, the most abundant species of *Carcharhinus* in this study, *C. leucas*, presently lives in coastal areas of tropical and subtropical riverine and lacustrine environments (*Compagno, 1984*). The second-most abundant species in this study, *Carcharodon carcharias*, inhabits inshore shallow water to open ocean and, as a top predator, feeds on larger marine mammals and fishes (*Ebert et al., 2013*; *Compagno, 2002*). While pelagic sharks *Carcharhinus plumbeus* and *C. longimanus* are also represented in this Pleistocene assemblage, the occurrences of *C. altimus*, *Aetobatus* sp., and *Myliobatis* sp. may suggest the possible presence of deeper sandy, flat bottoms (*Compagno, 1984*). The abundant associated marine vertebrate fossils, including teleost bones (*Tao, 1993*), otoliths (*Lin et al., 2018*), and whale bones (*Xue, 2004*), indicate a rich, thriving marine ecosystem in the area. The sedimentary environment of the Liuchungchi

Formation further points to shoreface to inner offshore setting, with several transgressive and regressive cycles (*Chen, 2016*). Taken together, the coastal areas in southwest Taiwan during the early Pleistocene can be interpreted as an inshore to offshore shallow-water environment with sandy bottoms.

## CONCLUSIONS

The fossil elasmobranch fauna from the tropical-subtropical West Pacific is poorly known compared to its modern analog, impeding our understanding of the formation of this current marine biodiversity hotspot. Using elasmobranch fossils from an early Pleistocene locality in southern Taiwan, we report a highly diverse shark and ray fauna from the region. The taxonomic composition of the assemblage reveals a nearshore shallow-water paleoenvironment which supports the sedimentary interpretation. In addition, the presence of †*Hemipristis serra* and large specimens of *Carcharodon carcharias* highlight the potential for studying fossils from underrepresented regions and stimulate similar studies from associated strata and localities. The present study can be regarded as the most extensive documentation on elasmobranch fossils from Taiwan.

## ACKNOWLEDGEMENTS

We would like to express our sincere gratitude to Prof. Hsi-Jen Tao (National Taiwan University) who donated the specimens to the Biodiversity Research Museum, Academia Sinica, Taiwan. We also thank Mrs. Hsiao I-Ju (Chiayi Municipal Museum, CMM) for her administrative assistance in examining the CMM collection, and Miss Sun You-Yu (National Taiwan Museum) for accessing the collection described by *Uyeno (1978)*. This manuscript has been improved based on constructive reviews by Kenneth De Baets, Dana Ehret, Laszlo Kocsis, and an anonymous reviewer.

### Funding

This work was supported by the Palaeontological Association, Stan Wood Grant Award (No. PA-SW202102) and the Ministry of Science and Technology, Taiwan (No. 109-2116-M-001-022-, 110-2116-M-001-009-) and Academia Sinica, Taipei, Taiwan, to Chien-Hsiang Lin. The funders had no role in study design, data collection and analysis, decision to publish, or preparation of the manuscript.

### Grant Disclosures

The following grant information was disclosed by the authors:
Palaeontological Association, Stan Wood Grant Award: PA-SW202102.
Ministry of Science and Technology, Taiwan: 109-2116-M-001-022, 110-2116-M-001-009.

### Competing Interests

The authors declare there are no competing interests.

## Author Contributions

- Chia-Yen Lin conceived and designed the experiments, performed the experiments, analyzed the data, prepared figures and/or tables, authored or reviewed drafts of the article, and approved the final draft.
- Chien-Hsiang Lin conceived and designed the experiments, performed the experiments, analyzed the data, authored or reviewed drafts of the article, and approved the final draft.
- Kenshu Shimada analyzed the data, authored or reviewed drafts of the article, and approved the final draft.

## Data Availability

The location of all specimens are deposited in the Chiayi Municipal Museum, Chiayi City, Taiwan (CMM), Biodiversity Research Museum, Academia Sinica, Taipei, Taiwan (BRMAS), and the National Taiwan Museum (NTM), respectively (see Introduction).

The accession numbers for each specimen are listed directly under "Referred specimens" in the "Systematic Paleontology". The summary of the examined specimens is available in Table 1.

## Supplemental Information

Supplemental information for this article can be found online at http://dx.doi.org/10.7717/peerj.14190#supplemental-information.

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
