# Peer review of "A previously overlooked, highly diverse early Pleistocene elasmobranch assemblage from southern Taiwan"

_PeerJ, doi:10.7717/peerj.14190_

## Round 0.1 · original submission · Major Revisions

You describe a diverse fauna of elasmobranchs from the Pleistocene of Taiwan, which fills an important research gap. All reviewers unanimously agree on this point. I would love to see this manuscript published, but there are some crucial points that need to be resolved before publication. The main point being:

Comparisons with other relevant localities: I agree with reviewer 2 that additional comparisons with Plio-Pleistocene of the Indo-West Pacific and Indo-Australian Archipelago would be relevant and needed.

Reworking: Many of your teeth are pristinely preserved as you highlight in the text, but there are also some exceptions that are very heavily worn. This makes it plausible that these teeth have been transported and reworked. Wear also hampers the robust taxonomic assignment of these teeth, which needs to be considered when assigning such teeth. It concerns particularly Alopias, Negaprion and Hemipristis (compare reviewer 2). Reworking should ideally be tested geochemically (Sr-Isotopy; rare earth elements; compare Reviewer 2) and would be highly recommended particularly for those taxa where an extension of stratigraphic and/or geographic range could be supported (e.g., Hemipristis). At least, the possibility should be discussed, and their stratigraphic range changed appropriately. Also, the impact of wear on their assignment should be more clearly discussed and considered by the use of open nomenclature.

Methodology: Some additional information on the sampling methodology (Which sieve mesh size was used? Why could otoliths be recovered but no shark teeth?) and stratigraphic assignment of the material (How can you be sure all material can be assigned to the Pleistocene with certainty? How can you rule out reworking/transport for the more worn teeth?) is needed for the sake of scientific reproducibility and comparability (compare reviewers 1 and 2). Also, additional information on size estimation of white sharks is somehow expected particularly as it is highlighted in the abstract (compare Reviewer 1 and Reviewer 2).

Taxonomic assignment and descriptions: In addition, to some of the very worn tooth already mentioned under the previous point (e.g., Alopias), there are some additional taxonomic assignments (e.g., Aetobatus, Dasyatis, Galeocerdo, Hemipristis) which need some additional scrutiny and/or comparisons with additional taxa or their intraspecific variation depending on their position (compare Reviewer 2 and Reviewer 3). In addition, some descriptions could be made more streamlined (avoiding repetition, reference to previous work concerning tooth homology; compare Reviewer 1).

Quantitative diversity analyses: I agree with Reviewer 2 that given the high diversity – even when questionable assignments are excluded – would merit the calculation of quantitative indices of diversity. Comparing the result with those of other regions from higher or lower latitudes would make your interpretations even more robust and even broader relevance.

Formatting: Please resolve some minor issues in grammar, spelling, and punctuation as well as discrepancies between references cited in text and reference lists (see particularly suggestions by Reviewer 1).

I consider most of these points minor but some are more major to moderate as they would merit additional analyses (reworking and diversity metrics).

In addition to these points, please make sure to address all other points raised by the reviewers and myself including those in annotated pdfs.

I look forward to receiving your revised manuscript.

·

Basic reporting

This manuscript presents novel information in a clear and concise manner. I have read through and annotated the manuscript making suggestions with regards to grammar, spelling, punctuation, etc. It is written in clear English however, some of the descriptions of specimens/taxa could be refined to be somewhat clearer. I have tried to offer suggestions in some cases to cut down on repetition and unclear descriptions (as an example, line 487 references the crown apex pointing apically. The apex is defined as the tip of the tooth, so obviously the tip of the crown would be pointing towards the tip of the tooth).Shimada's 2002 paper on dental homologies in the Lamniform sharks (cited in the manuscript) could help cut down on some of this repetition. The literature references are sufficient. I cross-checked the in-text cited literature with the references and noticed a few were not listed in the appropriate section and noted those cases. The authors are up-to-date (citing literature as recent as 2022) while also including classic references. The article is structured and formatted properly. I do not think any changes need to be made in that respect. The figures and tables are of a high quality and are all necessary and appropriate for the results of this study. Figure 2 does have a misspelling and I did have a few questions regarding the key (included in attached review). Aside from minor grammatical issues and the mentioned references in text that were omitted in the references, I think this manuscript should be accepted for publication. Therefore, I suggest acceptance with minor revisions.

Experimental design

The authors of this manuscript are describing a previously overlooked fauna of fossil shark teeth recovered from Taiwan. They review the geology and location of the finds, the methods by which the specimens were collected and review the species identified. The authors field collected specimens (both macro and micro) and also reviewed museum collections to gain a comprehensive review of the fauna. I believe the authors did their due diligence to sample the fauna. The introduction succinctly explains what the authors are planning to do and they do a good job of fulfilling what they set out to do. The methods could use some clarification. First, the authors discuss sieving 830 kg of sediment but they do not discuss what type of equipment was used. Nested screens? What size screen mesh was used? They mention the collection of otoliths so I assume it was a fine mesh, but the lack of shark teeth is a bit surprising. Secondly, the authors estimate the body length of a white shark based on one of the referred specimens. While this is an accepted method, they do not go into detail as to which tooth position they assigned the specimen to or how they came to that conclusion. This would only require the addition of a few sentences to the manuscript. Otherwise, I have no other issues with the design of this manuscript.

Validity of the findings

I think this manuscript is an important contribution to the knowledge of fossil chondrichthyans from Taiwan, which have been largely overlooked as the authors point out. I believe the identifications by the authors are accurate and their findings are important to document. I was expecting more of a discussion on the size of white sharks based on specimens because it was mentioned in the abstract. However the discussion of the topic was fairly brief. The authors utilized multiple museum collections as well as doing their own field work to gather a robust sample size.

Additional comments

I think this manuscript is a good contribution that should be published to document the occurrences of these taxa.

·

Basic reporting

The manuscript is clearly written and the reader can follow the science easily behind. As the paper is taxonomy focused, it has a specific structure related to the description of the reported species (e.g., synonym lists, referred materials, etc. see Systematic Paleontology chapter) that is appropriate for such investigation. There are lots of associated illustrations, that are high quality, and most are necessary.

The topic is properly introduced and overly focuses on the Taiwanese fossil record. However, the authors point out the importance of studying and comparing their finds to other fossil records from the tropical—subtropical region of the Indo-West Pacific (Lines 45-46). Here many other papers need to be cited, and also be used for taxonomic comparison. A compilation of the Indo-Australian Archipelago (IAA) elasmobranch record can be found in Kocsis et al. (2019). Also, relevant Plio-Pleistocene finds in the IAA should be considered such as from Sulawesi (Hoojier 1954) or Java (Yudha et al. 2018).

Experimental design

Methods of the study is properly stated. Most of the studied elasmobranch materials come from museum collections, however, interestingly hardly any elasmobranch teeth were recovered during fieldworks and screen-washing of five years intensive research (lines 104-106). This discrepancy calls for further explanation of the origin of the fossils.

The main scope of the paper is to report the overall Pleistocene diversity of elasmobranchs in Taiwan, and to place the fauna in a spatio-temporal context with neighboring high diversity regions. As mentioned above, the spatio-temporal comparison could be extended towards the tropical regions. On the other hand, the authors are taking granted that their whole fauna is Pleistocene in origin.

Some of the teeth are quite worn and clearly have been reworked and transported. Would it be possible that some of the teeth are originated from older beds? How the authors can be sure that the fossil assemblage is not mixed, and it contains only Pleistocene elements? Based on the geology and geography of the investigated area (Fig 1), the rivers higher upstream cut through older sediments such as Pliocene-Miocene. From these beds, older elasmobranch remains might have been reworked and mixed with younger material. Also, the large variation of sea level during the Pleistocene may also have resulted in eroding older rocks by the sea, and material from these may have re-deposited and mixed with the younger ones. The possible mixed nature of the material should be rigorously tested before the whole fauna is handled as Pleistocene. For example, Sr-isotope composition of few selected teeth, and also otoliths recovered from the proposed Pleistocene source beds could be analyzed, and the obtained data could help to test if wider range of mixing occurred. Other complementary approach, might be using trace element chemistry (e.g., rare earth elements) of the bioapatite material, which may further give hints in such taphonomic questions. See examples for such studies in Borneo where late Miocene material found reworked in modern coastal setting (e.g., Kocsis et al., 2021, 2022).

Validity of the findings

The major part of the paper is the taxonomic description of the fossils, and the reported exceptional high diversity. The fauna described here is very impressive, especially the large number of different carcharhinid species. There are, however, few reported taxa whose validity is questionable based on the figured material, and much stronger arguments are needed to accept them.

Alopias cf. vulpinus: one very worn tooth (Fig 3 C). In my opinion even the genus can be questionable. Based on one tooth with such preservation, I would be much more careful adding this taxon. Do you have more material? Anything that would prove further your identification? What about, for example, if it is a lateral Carcharias taurus tooth with the cupslets all broken and eroded away?

Carcharodon carcharias: these teeth are amazing and very important finds of the Taiwanese record. The size estimates of the sharks (total length - TL) based on the teeth’s size should be explained more in details in the view of more recent papers, and related different approaches (e.g., Shimada, 2019; Perez et al., 2021). The different calculations would also yield an error of TL estimates, which should be reported in the paper.

Hemipristis: Separating the fossil and the modern species need more explanation. Based on the sole tooth you show for H. elongata, (Fig 6 A), it could be a lateral tooth of H. serra. Please take a look at some of the regional, older occurrence of the species H. serra (e.g., Kocsis et al 2019: Figure 3 E-F). Those teeth are identical of that of yours. Would it mean that H. elongata existed in the late Miocene? Most probably not. Based on parsimony, the simplest explanation for this sole tooth in your record is to place it within H. serra. Of course, if there is more material, and modern comparative study then this can be further pursued. You may also take a look at the paper of Chandler et al. (2006).

In the manuscript, there is larger emphasizes on the presence of Hemipristis in the Pleistocene record, and even in the abstract it occurs as the first Pleistocene presence from the region. In fact, if you consider a bit wider region Hooijer (1954) has already mentioned this species from Pleistocene deposits of Sulawesi (Celebes). There might be also younger presence of the taxon in Java (Yudha et al. 2018, though this paper contains many mis-identifications). A further thought: this taxon could be specifically targeted with geochemical analyses (e.g., Sr-isotope analyses) in order to prove further their Pleistocene origin.

Carcharhinidae: many carcharhinids mentioned in details from north Borneo that you may consider adding to your spatio-temporal comparison (Kocsis et al 2019).

Carcharhinus amboinensis Line 299: see the separation of the two species based on finds in Borneo’s fossil record (Kocsis et al 2019).

Carcharhinus limbatus: Some of the teeth also resemble that of C. amblyrhynchoides (see Garrick, 1982), a shark that also exists in the region and around Taiwan (see Froese and Pauly, 2022).

Galeocerdo: close-up photo on the serration of the teeth should be shown in order to support the modern species G. cuvier opposite to that of fossil one: G. aduncus. (e.g., Cigala-Fulgosi and Mori 1979).

Negaprion acutidens: based on these worn teeth that represented by only main cups it is very brave to say even the genus (Fig 16 A-C). For instance, how do you discard the possibility that these teeth are not coming from genus Carcharhinus representing broken, worn lower teeth? To have this taxon in your record, much better specimens are needed to be shown.

Dasyatis: How do you discard the possibility of the genus Himantura for these teeth (Fig 18)? In my opinion, until you do not show a detailed comparison of the dentation of extant dasyatids from the region, you cannot go lower taxonomy than family level (i.e., Dasyatidae). Himantura might be an equally acceptable genus for these teeth. See new compilation on revised batoid taxonomy in Last et al. (2016a,b). It would be nice to see reported the number of different dasyatids that may occur in Taiwan’s waters today.

Aetobatus: Have you considered other species? A. flagellum? Have you compared the teeth of different Aetobatus species in the region? Maybe reporting these teeth as open nomenclature would be more appropriate.

Aetobatus-Myliobatus-Aetomyleus and maybe Rhinoptera may be clarified here. Please, if you go down to species level with these myliobatiformes, then the modern species’ teeth should be shown as comparison. Also, the connections-interlocking between the different tooth plates seems to be an important factor among these rays (e.g., Hovestadt & Hovestadt-Euler 2013). As for the sharks, the number of ray species of modern taxa occurring in the region should be added to the Remarks, in order to have a clearer picture of modern diversity and the possibility of other species that may have similar teeth.

Diversity: The claimed high diversity would still be impressive if the questionable taxa are taken out (e.g., Alopias, Negaprion, H. elongata). What is missing is the quantitative expression of diversity, for example using diversity indices (e.g., Shannon, Fisher alpha), and comparing these indices with other faunas from more temperate regions to the north (e.g., Japan) and more tropical ones (e.g., Borneo, Celebes IAA) to the south of Taiwan. This kind of data would further raise the quality of the paper.

Additional comments

Summary: The presented elasmobranch taxonomical report from Taiwan is very intriguing and it fills an important gap about our knowledge of how past biodiversity has changed and varied in the region. However, the work needs (1) important clarification of the possible mixed nature of the fauna (Pleistocene vs older elements), (2) more inputs on spatio-temporal comparison in the region (e.g., tropical, high diversity regions), (3) amending taxonomy, and (4) presenting quantitative diversity data. Then the paper could be a great contribution to PeerJ, and expected to be highly cited. Based on all the comments above, here a major/moderate revision is suggested.

Mentioned References under points 1 to 3:

Chandler, R.E., Chiswell, K.E., Faulkner, G.D. 2006. Quantifying a Possible Miocene Phyletic Change in Hemipristis (Chondrichthyes) Teeth. Palaeontologia Electronica Vol. 9, Issue 1; 4A:14p

Cigala-Fulgosi, F. & Mori, D. 1979. Osservazioni tassonomiche sul genere Galeocerdo (Selachii, Carcharhinidae) con particolare riferimento a Galeocerdo cuvieri (Peron & Lesueur) nel Pliocene del Mediterraneo. Bollettino della Societa Paleontologica Italiana, 18, 117–132.

Froese, R. and D. Pauly. Editors. 2022. FishBase. World Wide Web electronic publication. www.fishbase.org, version (02/2022).

Garrick, J. A. F. 1982. Sharks of the genus Carcharhinus. NOAA Technical Report NMFS Circular vol. 445, NMFS Scientific Publications Office, Seattle, USA, 194 pp.

Hooijer, D. 1954. Pleistocene vertebrates from Celebes. IX. Elasmobranchii. X. Testudinata. Proceedings of the Koninklijke Nederlandse Akademie van Wetenschappen, Series B, 57, 475–789.

Hovestadt, D.C., Hovestadt-Euler, M. 2013. Generic assessment and reallocation of Cenozoic Myliobatinae based on new information of tooth, tooth plate and caudal spine morphology of extant taxa. Palaeontos, 24, 1–66.

Kocsis, L., Razak, H., Briguglio, A., Szabó, M. 2019. First report on a diverse Neogene fossil cartilaginous fish fauna from Borneo (Ambug Hill, Brunei Darussalam). – Journal of Systematic Palaeontology, 17(10), 791–819. - https://doi.org/10.1080/14772019.2018.1468830

Kocsis, L., Botfalvai, G., Qamarina, Q., Razak, H., Király, E., Lugli, F., Wings, O., Lambertz, M., Raven, H., Briguglio, A., Rabi, M. 2021. Geochemical analyses suggest stratigraphic origin and late Miocene age of reworked vertebrate remains from Penanjong Beach in Brunei Darussalam (Borneo). – Historical Biology, 33/11, 2627–2638.

Kocsis, L., Briguglio, A., Cipriani, A., Frijia, G., Vennemann., T., Baumgartner, C., Roslim, A., (2022): Strontium isotope stratigraphy of late Cenozoic fossiliferous marine deposits in North Borneo (Brunei, and Sarawak, Malaysia) – Journal of Asian Earth Sciences, doi.org/10.1016/j.jseaes.2022.105213.

Last, P. R., Naylor, G. J. P., Manjaji-Matsumoto, B. M. 2016a. A revised classification of the family Dasyatidae (Chondrichthyes: Myliobatiformes) based on new morphological and molecular insights. Zootaxa, 4139, 345–368.

Last, P. R., White, W. T., de Carvalho, M. R., Séret, B., Stehmann, M. F. W., Naylor, G. J. P. 2016b. Rays of the world. CSIRO Australian National Fish Collection, Hobart, 800 pp.

Perez, V.J., Leder, R.M., Badaut, T. 2021. Body length estimation of Neogene macrophagous lamniform sharks (Carcharodon and Otodus) derived from associated fossil dentitions. Palaeontologia Electronica, 24(1):a09. https://doi.org/10.26879/1140palaeo-electronica.org/content/2021/3284-estimating-lamniform-body-size

Shimada, K. 2019. The size of the megatooth shark, Otodus megalodon (Lamniformes: Otodontidae), revisited. Historical Biology, 1–8.

Yudha, D.S., Ramadhani, R., Suriyanto, R.A., Novian, M.I. 2018. The diversity of sharks fossils in Plio-Pleistocene of Java, Indonesia. AIP Conference Proceedings 2002 (1), 020013: doi.org/10.1063/1.5050109

Reviewer 3 ·

Basic reporting

This manuscript is clear, concise and well-written. Professional English is used throught.

The references are appropriate to this study and are cited in logical places within the manuscript. Cited references include some of the most recently published papers relevant to this subject.

The overall structure is very professional and well-organized. In particular, the photographs of individual teeth are very crisp and detailed.

Experimental design

The experimental design of this manuscript is based on surface collections of specimens housed in three museums. They acknowledge that surface collecting will produce a size-related bias against smaller teeth. But their methodologies are consistent with other published research on fossil elasmobranchs and allows comparisons with these assemblages.

Validity of the findings

The authors have done an excellent job of documenting this fauna, particularly the difficult genus Carcharhinus. This genus is particularly diverse during the Neogene and Quaternary with many species having similar tooth morphologies. Sorting these species out in a uniform and manageable manner is admirable.

Three specific suggestions for the manuscript:

• The tooth identified as Alopias cf. vulpinus is not consistent with Alopias tooth morphology. The root shape and absence of an enameloid overhang on the labial face of the root are problematical. The authors attribute the absence of these to taphonomic loss. This is possible, particularly since the dull, patchy enameloid is consistent with acid-etching, such as would occur when a tooth passes through the gastrointestinal tract. But the extensive, symmetrical erosion of the root lobes seems implausible. Given the characteristics of this tooth, an identification as a lower distal tooth of a larger lamniform (e.g., Isurus oxyrinchus) with shorter root lobes seems more likely.

• The tooth identified as Hemipristis elongatus is likely from a very distal position in the dentition as indicated by the straight basal root margin. Identifying the species from such a distal tooth is difficult. A review of the information in Chandler et al. 2006. Quantifying a Possible Miocene Phyletic Change in Hemipristis (Chondrichthyes) Teeth. Palaeontologia Electronica Vol. 9, Issue 1; 4A:14p would be helpful in assessing this tooth.

• Lower teeth are described for both Carcharhinus leucas and C. longimanus. These lowers are unlike those of other Carcharhinus in the assemblage, but quite similar to each other. A discussion of how to distinguish the lower teeth of these two species would be useful. The ‘Remarks’ section of C. leucas would be an appropriate place for this comparison.

Additional comments

Overall, a very clean and well-structured documentation of this poorly-known fauna.

---

## Round 0.2 · Minor Revisions

Thank you for taking the time to thoroughly address the reviewer comments. The integration of diversity indices and comparisons with other Pleistocene sites has made the manuscript easier to follow and even wider importance. Your paper is close to acceptance but there remain a few minor but crucial points which need to be addressed before publication. The main points being:

Diversity analyses: The added diversity analyses really make your arguments stronger. However, additional background information concerning calculation of diversity metrics is necessary. Additional discussion on high diversity is also warranted (see comments by reviewer 1). Particularly a discussion to why high species diversity is important and ideally the potential mechanisms behind it.

Reworking: the added discussion on possible reworking is very welcome. I agree with reviewer 2 that you cannot entirely rule out the inclusion of potentially reworked material (particularly the more poorly preserved/worn material). In this respect, I would like you to add a small added sentence that you cannot rule entirely that certain worn/rare material (please list which taxa this would concern) – see previous decision/reviews) is not reworked although that you consider this implausible (listing the main arguments against it)

Grammatical/typographical issues: there remain some minor grammatical and typographical issues which need to be resolved (compare reviewer 1 and 2).

Please address these and all other points (including those in annotated pdfs) raised by the reviewers.

I look forward to receiving the revised manuscript and the publication of this work.

·

Basic reporting

Article is much improved on re-review. I made some minor comments regarding grammar but otherwise the authors have fulfilled all the requirements.

Experimental design

The authors added a table calculating the diversity indices of multiple Pleistocene elasmobranch assemblages are requested by one of the reviewers. I think this illustrates the argument the author's were making, that their locality is a very diverse assemblage. However, there is nothing in the materials and methods about calculating the indices and they are mentioned in the discussion, however the authors do not discuss much about the high diversity. I felt like there was more discussion about the diversity in the earlier draft without data, and now they have the data to back up the claims but do not discuss it.

Validity of the findings

No comment, I think the authors' findings are meaningful and valid.

Additional comments

I think this article is much improved. I have attached an annotated draft of the manuscript with suggestions for grammatical changes. My only comments are related to the discussion section. The last sentence in paragraph 3 (line 630-631) insinuates that this assemblage is important due to the high diversity, but the authors do not discuss what that importance is, or how this assemblage is important. This goes back to my comments under Experimental design. The authors show that species diversity is very high in this assemblage but do not explain why this is important. Similarly, the second to last sentence in the Conclusion (lines 696-699) tease the presence of H. serra and large C. carcharias as being important but do not actually tell us why. Otherwise, I thought the manuscript was much improved!

·

Basic reporting

The authors followed most of the recommendations from the previous review and referred to the raised issues. The manuscript is noticeably improved, and the taxonomic work is nicely clarified. The added diversity indices and comparison with other Pleistocene sites further enhance the value of this work. Therefore, the result of this research will be a great contribution to PeerJ. I have only few minor remarks, otherwise the paper is suggested for publication.

Experimental design

The manuscript meets the standards. See additional comments.

Validity of the findings

(1) On the reworking topic: The authors state at lines 121-122 that the lithologically distinct older sedimentary blocks are below their sampling site. However, the older rocks of Miocene age crop out 1 km upstream (Line117). I cannot see why older material could not be transported down and be mixed with younger ones. Yes, the authors did not sample these older blocks, but the in-situ Pleistocene rocks did not yield much teeth as well. Moreover, if reworking happened during the Pleistocene so in the “bone-teeth beds”, that may have been sampled by others for the museum collections, still could have contained some older materials. So, stating improbable mixing and that the entire assemblage originated from the Pleistocene (line 123-125), is a bit overlooking the geological situation. Nevertheless, it is understood that any geochemical approaches to clarify the reworking issue is out of the scope of this study, but maybe a sentence mentioning that some of the worn specimens may have somewhat older origin could be added here (lines ~123-125). Especially, when the authors mention that good preservation of Hemipristis teeth rules out origin from older beds (Lines 668-669), so would not this mean that bad preservation may have older origin? I think there might be a slight chance.

Additional comments

(2) Lines 641-648: TL estimates of C. carcharias is now indeed more detailed. It would have been nice to see the TL range with the method of Perez et al 2021, just for the sake of comparison. Not a must, but it is somewhat different approach than that of Shimada’s work.

(3) Line 665: typo Hooiher --> Hooijer

Congratulation for this excellent work. I hope to see it soon published,
All the best,
Laszlo Kocsis

Reviewer 3 ·

Basic reporting

No comment.

Experimental design

No comment.

Validity of the findings

No comment.

Additional comments

The resubmitted manuscript has addressed my concerns/comments from the initial review. I have no further issues with this manuscript.

---

## Round 0.3 · accepted · Accept

Thank you for addressing and clarifying these final suggestions which have made the manuscript easier to follow and of even greater value. It has been a pleasure to handle your manuscript and I look forward to seeing it published. Reviewer 1 still raised some minor grammatical issues which I would like you to address during the proofing phase.

·

Basic reporting

The article is clear and concise. In the attached document I noted a number of small grammatical issues that can be found on lines: 40, 66, 105, 115, 190, 195, 282-283, 423, 446, 613, 615, 638 and 640. Most of these are small issues that could be fixed and some could be left as is (depending on the author's opinion). Otherwise, I think the data, references and images are all of high quality.

Experimental design

Through revision, this article meets all the requirements of PeerJ when it comes to methods, standards and research.

Validity of the findings

The authors have done due diligence to properly identify species and the geology. They have fleshed out the portions of the manuscript previously commented on by the reviewers and I am satisfied with their changes.

Additional comments

See above comments regarding the grammatical issues that I have highlighted in the manuscript. On line 615 the authors use the genus 'Isurus' for Carcharodon hastalis. When the citing literature was published that genus was being used, however it was changed to Carcharodon in Ehret et al., 2012 (which the authors cite in the paper). I do understand however if the authors are direct quoting the literature. I do not need to review this manuscript again, I am very happy with the edits made and the authors' reasoning for making changes.

·

Basic reporting

The manuscript has been further amended, and the authors' explanations to the raised issues are acceptable. The work is ready to be published.

Experimental design

No further comments

Validity of the findings

No further comments.

Additional comments

No further comments.